# Quantifying the potential benefits of risk-mitigation strategies on future flood losses in Kathmandu Valley, Nepal

Carlos Mesta[1], Gemma Cremen[2], Carmine Galasso[1,2]

[1]Understanding and Managing Extremes (UME) Graduate School, Scuola Universitaria Superiore (IUSS) Pavia, Pavia, 27100, Italy

[2]Department of Civil, Environmental and Geomatic Engineering, University College London, London, WC1E 6BT J, United Kingdom

*Correspondence to*: Carmine Galasso (c.galasso@ucl.ac.uk)

**Abstract.** Flood risk is expected to increase in many regions worldwide due to rapid urbanization and climate change if adequate risk-mitigation (or climate-change-adaptation) measures are not implemented. However, the exact benefits of these measures remain unknown or inadequately quantified for potential future events in some flood-prone areas such as Kathmandu Valley, Nepal, which this paper addresses. This study examines the present (2021) and future (2031) flood risk in Kathmandu Valley, considering two flood-occurrence cases (with 100-year and 1000-year mean return periods) and using four residential exposure inventories representing the current urban system (Scenario A) or near-future development trajectories (Scenarios B, C, D) that Kathmandu Valley could experience. The findings reveal substantial mean absolute financial losses (€ 473 million and € 775 million in repair/reconstruction costs) and mean loss ratios (2.8% and 4.5%) for the respective flood-occurrence cases in current times if the building stock's quality is assumed to have remained the same as in 2011 (Scenario A). Under a "no change" pathway for 2031 (Scenario B), where the vulnerability of the expanding building stock remains the same as in 2011, mean absolute financial losses would increase by 14%-16% over those of Scenario A. However, a minimum (0.20 m) elevation of existing residential buildings located in the floodplains and the implementation of flood-hazard-informed land-use planning for 2031 (Scenario C) could decrease the mean absolute financial losses of the flooding occurrences by 9%-13% and the corresponding mean loss ratios by 23%-27%, relative to those of Scenario A. Moreover, an additional improvement of the building stock's vulnerability that accounts for the multi-hazard-prone nature of the valley (by means of structural retrofitting and building code enforcement) for 2031 (Scenario D) would further decrease the mean loss ratios by 24%-28% relative to those of Scenario A. The largest mean loss ratios computed in the four scenarios are consistently associated with populations of the highest incomes, which are largely located in the floodplains. In contrast, the most significant benefits of risk mitigation (i.e., largest reduction in mean absolute financial losses or mean loss ratios between scenarios) are experienced by populations of the lowest incomes. This paper's main findings can inform decision makers about the benefits of investing in forward-looking multi-hazard risk-mitigation efforts.

## 1 Introduction

Flooding is among the world's most prevalent natural hazards (World Meteorological Organization, 2021). Across the world, tens of millions of people are displaced from their homes by flooding each year, while related damage to property and physical infrastructure causes hundreds of billions of U.S. dollars in direct losses (e.g., Hallegatte et al., 2017; IDMC, 2015). For instance, in the United States alone, the national flood-induced average annual losses (AAL) for 2020 were approximately USD 32.1 billion, with the most impoverished communities across the nation experiencing the largest values of this metric normalized on the basis of building replacement cost (Wing et al., 2022). In general, it is estimated that four of every ten people exposed to flood risk globally live in poverty (Rentschler et al., 2022), which means that the human impacts of flooding tend to be concentrated disproportionately among low-income communities and countries. No matter how frequent or small, flood events can disrupt years of development and poverty reduction progress in these countries (Hallegatte et al., 2016).

Moreover, flood risk is expected to increase due to climate change impacts (e.g., intensification of rainfall extremes, sea level rise) and socioeconomic development in flood-prone regions (e.g., Ceola et al., 2014; Hirabayashi et al., 2013; Jongman et al., 2012; Nicholls et al., 2021). Specifically, rapid urbanization – which is expected to mainly feature across cities in Asia and Africa over the next few decades (United Nations, 2019a) – could increase flood exposure and vulnerability (e.g., Hemmati et al., 2020) and intensify flood hazard (by increasing runoff during precipitation events, due to the replacement of natural ground with impermeable surfaces, changes to drainage or irrigation systems, and deforestation, for instance), if not correctly managed. Therefore, there is an urgent need to investigate how cities can effectively adapt to dynamic risk challenges, especially in low-income regions (Cremen et al., 2022b; Fraser et al., 2016; Hinkel et al., 2018; Jongman, 2018).

Nepal is a landlocked country in South Asia, located in the Himalayan region. Its complex topography and social and physical exposure and vulnerability make Nepal particularly susceptible to geological (e.g., earthquakes, landslides) and hydro-meteorological hazards (e.g., floods, droughts). According to the Global Climate Risk Index, Nepal was among the top ten countries most affected by extreme weather events over the 2000-2019 period (Eckstein et al., 2021). Flooding is the most frequent natural hazard affecting Nepal. Apart from fluvial (riverine) flooding during the monsoon season, other types of flooding that the country experiences include pluvial (flash) flooding from heavy rainfall in mountainous areas, glacial lake outburst flooding, and landslide-induced flooding (Landell Mills, 2019). Nepal has a long history of devastating flood events, such as those that occurred in the Eastern region (1987), Central Nepal (1993), and Kushaha (2008) (Adhikari, 2013; Government of Nepal, 2017). Monsoon precipitation across South Asia in August 2017 affected 80% of the Terai region and surrounding districts. Terai is one of Nepal's three ecological belts (together with Mountain, and Hill), and covers the alluvial and fertile plains along the southern part of the country (Government of Nepal, 2017). The resulting widespread flooding caused 160 deaths and 45 injuries, destroyed 41,626 houses, and partially damaged 150,510 houses. Direct losses were estimated to be USD 584.7 million, of which 32% corresponded to the housing sector (Government of Nepal, 2017).

The 2017 Terai flood and earlier major events have emphasized the significant risk that flooding continuously imposes on the Nepalese population. While flood risk is already substantial, several ongoing trends in the country could further amplify this risk in the coming years. Firstly, Nepal is projected to be one of the fastest urbanizing countries in the world over the 2018-2050 period (United Nations, 2019b), which could lead to significantly larger amounts of flood exposure. While urban growth is gaining pace across different regions of Nepal, Kathmandu Valley represents the "hub" of urban development in the country (Timsina et al., 2020). A previous study by the authors (Mesta et al., 2022b) revealed that urban land in Kathmandu Valley could reach 352 km$^2$ in 2050, almost doubling its current size and covering half the total valley extent. A significant share of this new urbanization is projected to occupy the valley's most hazardous (at least in terms of flooding and liquefaction) and socially-vulnerable regions (Mesta et al., 2022b). Secondly, other natural hazards such as earthquakes have unveiled the poor state of Nepal's building stock and physical infrastructure, which is caused by a combination of low-quality building materials, deficient construction practices, low compliance with building codes, as well as aging, and deterioration (Bothara et al., 2018; Varum et al., 2018). Traditional materials, such as bamboo/wood, stone, and mud, are still preferred in many regions of the country (especially in rural areas) due to their availability and low cost (Bothara et al., 2018). However, buildings made of bamboo/wood or mud suffer severely from flood damage (e.g., Becker Andrea B. et al., 2011; Fatemi et al., 2020) due to low durability and high permeability. Thirdly, climate change scenarios developed by the Government of Nepal (Ministry of Forests and Environment, 2019) reveal a rising trend in precipitation (for all seasons, except the pre-monsoon season) in the medium term (2016-2045) and long-term (2036-2065). Therefore, it is critical to determine the potential benefits of implementing disaster risk reduction (DRR) strategies in the country (particularly Kathmandu Valley) towards preventing devastating economic losses and casualties in future major natural hazard events.

Over the last few decades, several flood risk assessments have been conducted at various geographic scales, often leveraging the most recent high-resolution flood, assets, and population maps (Rentschler et al., 2022). However, most studies have focused on high-income countries (Chakraborty Jayajit et al., 2014; Oubennaceur et al., 2019), and studies for developing countries have mostly concentrated on large economic centers, such as Jakarta (Budiyono et al., 2015), Dhaka (Gain et al., 2015), or Ho Chin Minh City (Bangalore et al., 2019). Additionally, some researchers have examined how flood adaptation measures (e.g., ring dike, wet-proofing, dry-proofing, elevating roads and buildings) and/or urban development can affect flood risk trajectories (e.g., Chang et al., 2019; de Ruig et al., 2019; de Ruiter et al., 2021; Du et al., 2020; Lasage et al., 2014; Scussolini et al., 2017; Song et al., 2018; Thieken et al., 2016). Similar studies have yet to be conducted for Kathmandu Valley, however.

This study contributes to the efforts required to quantify the benefits of appropriate mitigation strategies on growing flood risk in urban areas for informing and promoting risk-sensitive decision-making (e.g., Cremen et al., 2022; Galasso et al., 2021a). The work explicitly investigates the effect of various risk-mitigation strategies (i.e., elevating buildings, flood-hazard-informed land-use planning, building retrofitting, and building-code enforcement) on flood-induced financial losses in Kathmandu Valley, Nepal. The methodology is a scenario-based flood loss estimation approach, using 100-year and 1000-year mean return

period flood occurrence maps and four exposure and vulnerability scenarios representing the current (2021) and potential near-future (2031) development trajectories for the valley, focusing only on residential buildings. Note that the impact of climate change is not explicitly considered within this work. The results can be relevant to various stakeholders, providing a clear quantitative description of the potential flood risk and its mitigation in Kathmandu Valley that can be leveraged for decision-making on investments in risk-reduction programs.

## 2 Study area

This study focuses on Kathmandu Valley, Nepal, which is surrounded by the Himalayan mountains and lies within the Bagmati river basin. The Bagmati river is 170 km in length, originates north of Kathmandu Valley at an altitude of 2690 m, and flows south through Nepal to reach the Ganges in India. Climatically, the Bagmati river basin can be divided into three regions: subtropical climate (elevations lower than 1000 m), warm temperate climate (elevations between 1000 m and 2000 m), and cool temperate climate (elevations higher than 2000 m; Dhital et al., 2013). The annual average and monsoon average rainfall of the catchment area are 1800 mm and 1500 mm respectively, and the mean temperature varies between 10°C and 30°C (Dhital et al., 2013).

Kathmandu Valley occupies a total area of 721 km$^2$ consisting of three districts (Bhaktapur, Kathmandu, and Lalitpur), which comprise five municipal areas and several municipalities and rural municipalities (formerly named village development committees, or VDCs). The built-up areas in Kathmandu Valley are estimated to be 202 km$^2$ for 2021 and are expected to increase to 307 km$^2$ by 2031 (Mesta et al., 2022b). Figure 1 provides a physical map of Kathmandu Valley, showing elevation (available at https://earthexplorer.usgs.gov/, last accessed December 2022) and the river network (available at https://openstreetmap.org/, last accessed December 2022). Figure 2 shows the administrative division of Kathmandu Valley and its built-up areas in 2021 and 2031 (Mesta et al., 2022b).

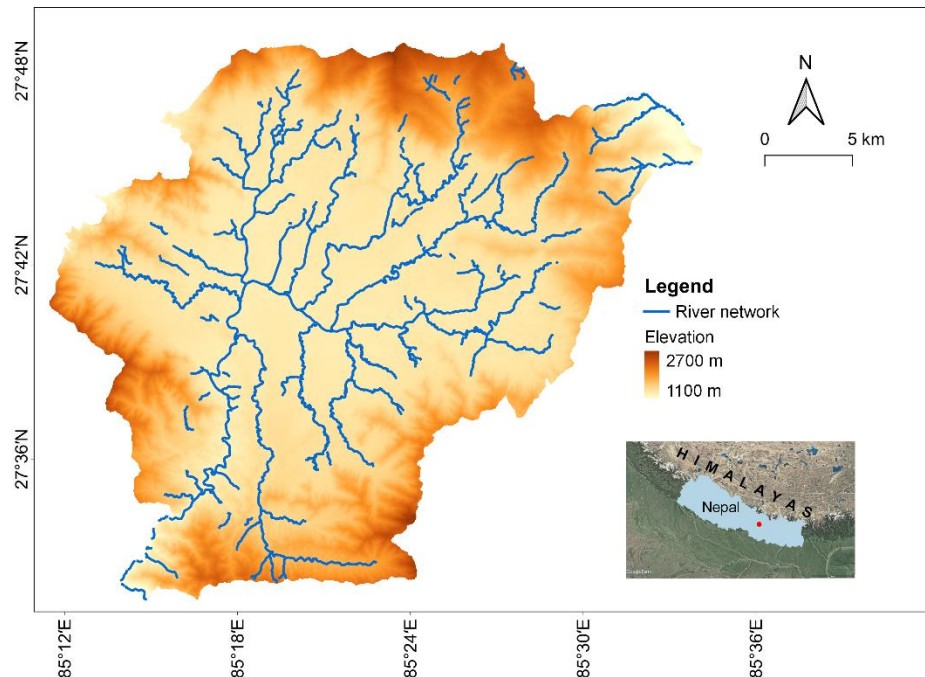

**Figure 1.** Physical map of Kathmandu Valley. The river network is taken directly from OpenStreetMap (OSM); small streams appear cut off where OSM data are incomplete. Inset map data: © Google Earth.

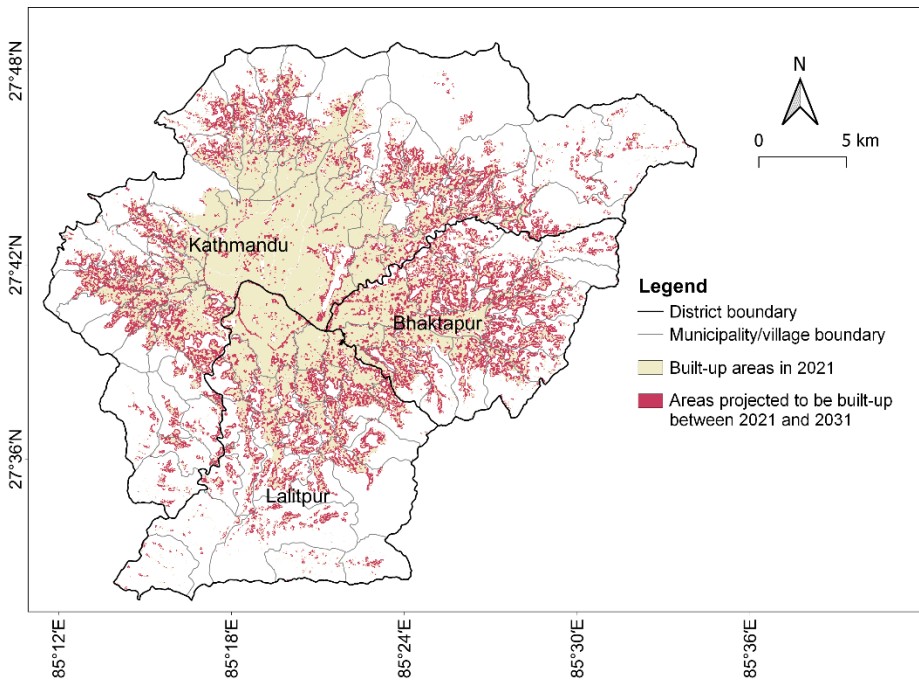

**Figure 2.** Administrative map of Kathmandu Valley and its built-up areas (Mesta et al., 2022b)

## 3 Materials and methods

Quantifying flood risk requires modeling hazard (flood extent and flood depths), exposure (locations and characteristics of population and buildings), and vulnerability (the extent to which hazard affects exposed assets). Figure 3 provides a scheme that summarizes the methodology implemented in this study. The following subsections present further details of the study area, as well as the methods and data used for the analysis.

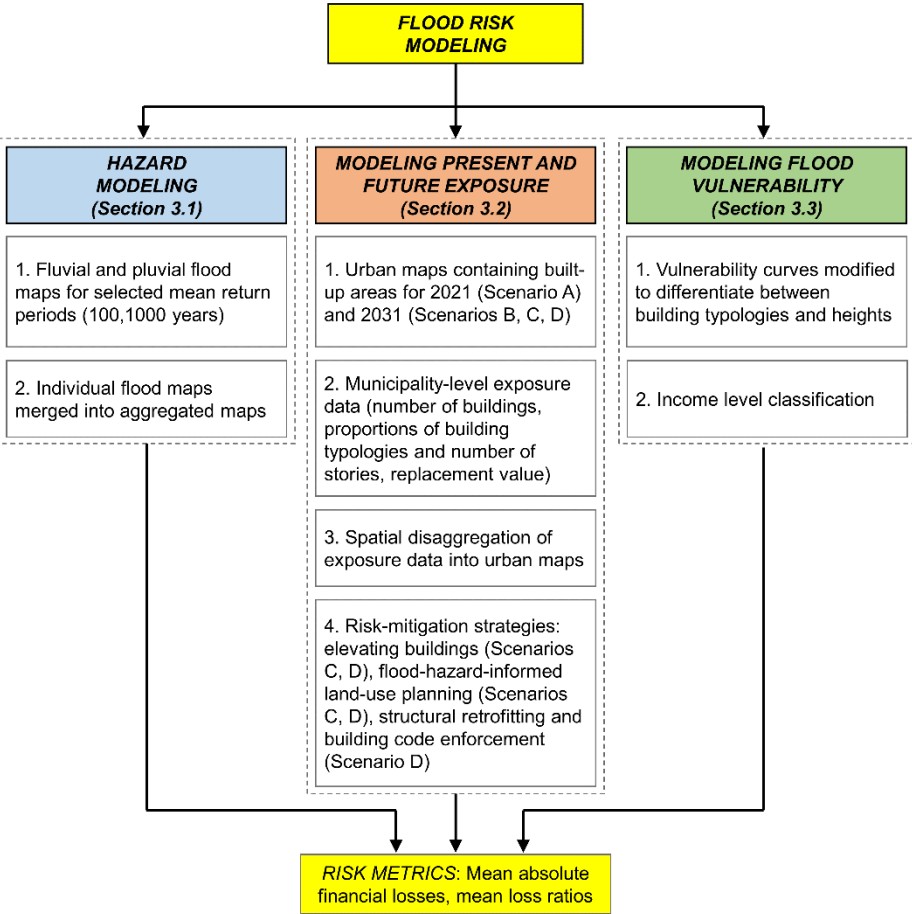

**Figure 3.** Overview of the flood risk modeling approach used in this study

### 3.1 Hazard modeling

We use the high-resolution Fathom-Global 2.0 model (Sampson et al., 2015), which accounts for both fluvial and pluvial inundation. This global model uses the Multi-Error-Removed Improved-Terrain (MERIT) digital elevation model (Yamazaki et al., 2017) and MERIT Hydro (Yamazaki et al., 2019) as topography and hydrography datasets, respectively. These data
provide the most accurate representation of ground surface elevation and location of the rivers at the global scale, which is critical to building robust flood models (Fathom, n.d.). Fluvial inundation is simulated in all river basins with upstream

catchment areas larger than 50 km$^2$. At the same time, pluvial flooding is captured for all catchment sizes by simulating rainfall directly onto the modeled topography. The model considers a 2D shallow-water formulation to explicitly simulate flood wave propagation and a regionalized flood frequency analysis (Smith et al., 2015) to derive river discharge. Fathom-Global provides 90 m-resolution maps of flood extents and flood depths for multiple mean return periods (from 1:5 year to 1:1000 year). Note that hazard data of finer resolutions (e.g., 10 m or lower) are recommended for capturing the highly localized nature of flood hazard (e.g., for representing small streams accurately) and associated risks at the urban scale (e.g., Afifi et al., 2019; Nofal and van de Lindt, 2021). In addition, urbanization effects on flood hazard (i.e., the replacement of natural ground with impermeable surfaces, changes to drainage or irrigation systems, and deforestation can increase runoff during precipitation events) are not explicitly accounted for by the Fathom-Global model and are therefore neglected in our analyses. However, the primary purpose of this study is to test different exposure/vulnerability scenarios using a common flood hazard input that is open and easily accessible; developing bespoke fine-resolution flood hazard models for the study area is not within the scope of this work.

We consider two cases of flooding occurrence in Kathmandu Valley. The first case is based on the Fathom-Global undefended flood map with a 100-year mean return period (i.e., 1% annual exceedance probability). Decision makers frequently use this type of map (e.g., to identify flood risk zones in the United States) (Ludy and Kondolf, 2012; Federal Emergency Management Agency (FEMA), 2010). The second flood-occurrence case reflects a situation in which flooding is more severe and is based on the Fathom-Global undefended flood map with a 1000-year mean return period. The flood maps are resampled to 30 m using the nearest neighbor method to match the spatial resolution of the exposure maps (Díaz-Pacheco et al., 2018). We combine individual flood maps into aggregated hazard maps that represent fluvial-pluvial flooding for each mean return period by taking their maximum depths in line with the method of Tate et al. (2021), who mosaiced fluvial and pluvial flood grids to generate an aggregated flood hazard map for the United States. The fluvial-pluvial hazard maps for each considered mean return period are presented in Figure 4; the individual flood maps are available online through the METEOR project (https://maps.meteor-project.org/map/flood-npl/, last accessed December 2022). Hereafter, we describe the flooding-occurrence cases using only the terms "100-year" and "1000-year", omitting the description "mean return period" for brevity. Overall, the aggregated flood maps are largely dominated by the effects of pluvial flooding : in both 100-year and 1000-year aggregated flood maps, around 15% of the flooded areas are exposed to both types of flooding, 84% are only exposed to pluvial flooding, and less than 1% are only exposed to fluvial flooding. It should be noted that fluvial flooding generally results in low-velocity flows dominated by hydrostatic pressure, while pluvial flooding often features higher flow velocities (Gentile et al., 2022); these differences in velocity characteristics could be important for estimating flood damage in areas with steep terrain (Nofal and van de Lindt, 2022). However, we use only flood depth as the intensity measure in this study, since it is widely used for flood loss estimation (e.g., Federal Emergency Management Agency (FEMA), 2022; Nofal and van de Lindt, 2022), and flood velocities are more difficult to record than flood depths, requiring hydraulic simulations (e.g., Kreibich et al., 2009).

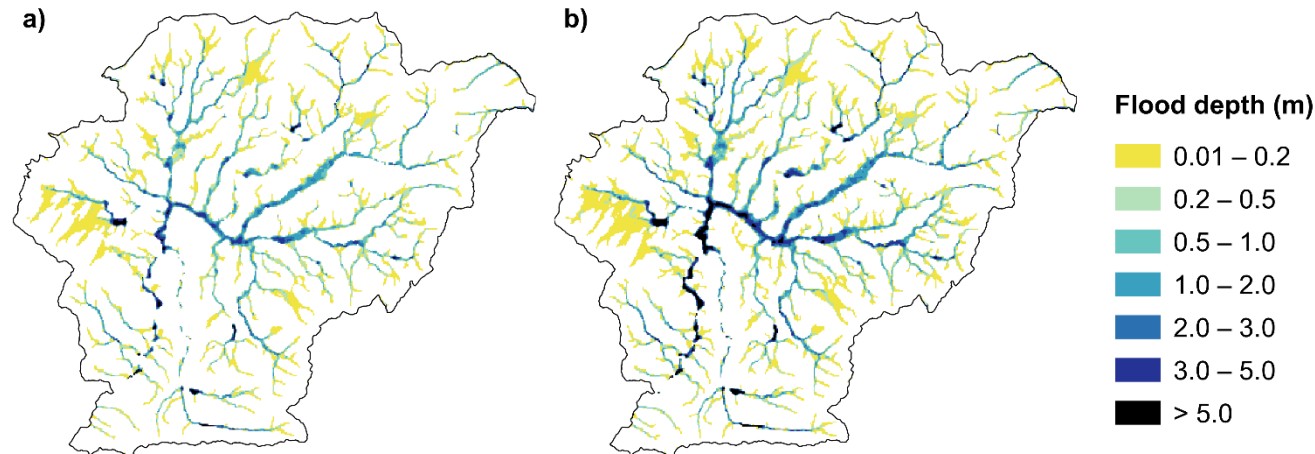


**Figure 4.** Fluvial-pluvial flood maps for a) 100-year mean return period; and b) 1000-year mean return period flooding occurrences. The individual flood maps are available online through the METEOR project (https://maps.meteor-project.org/map/flood-npl/, last accessed December 2022).

### 3.2 Modeling present and future exposure

We use the four exposure scenarios A-D for Kathmandu Valley developed in Mesta et al. (2022a), which are based on the National Population and Housing Census 2011 (Central Bureau of Statistics (CBS), 2012) and various assumptions on the estimated population and number of households for 2021 and 2031. Note that the Government of Nepal postponed the census planned for 2021 due to the COVID-19 pandemic, so the characterization of 2021 urban development relies on 10-year-old data.

The proposed scenarios portray different conditions for Kathmandu Valley in terms of urban growth, the prevalence of varying building typologies, and the implementation of DRR measures (see Table 1). Seven building typologies are included in the considered exposure scenarios: adobe (A), brick/stone masonry with mud mortar (BSM), brick/stone masonry with cement mortar (BSC), wood-frame (W), current-construction-practice reinforced concrete (RC-CCP), well-designed reinforced concrete (RC-WDS), and reinforced masonry (RM). These typologies have been previously used by Chaulagain et al. (2016,

2015) as well as Mesta et al. (2022a) to estimate seismic losses in Nepal and Kathmandu Valley. A, BSM, and BSC buildings constitute unreinforced masonry structures. RC-CCP refers to RC frame structures constructed without technical supervision. In contrast, RC-WDS are RC structures with ductile detailing designed according to seismic provisions. RM corresponds to the RM1 (Reinforced Masonry Bearing Walls with Wood or Metal Deck Diaphragms) building typology from the HAZUS Earthquake Model (Federal Emergency Management Agency (FEMA), 2020).

The specifications of each scenario are primarily provided in Mesta et al. (2022a); any deviations in these details for this specific study are documented in Sections 2.3.1 to 2.3.3. Note that we do not consider the presence of basements for any building typology since they do not seem to be a common feature of buildings in the valley (except for a minor proportion of

RC buildings). For instance, an extensive post-earthquake damage assessment conducted by the National Society for Earthquake Technology-Nepal (NSET, 2016) described the presence of basements in modern high-rise RC buildings; however,

buildings with six or more stories only represent 1% of all buildings in Kathmandu Valley. Suwal et al. (2017) identified the presence of basements in 31% of RC buildings in the valley, but their study was limited to only 64 buildings.

### 3.2.1 Scenario A (population and buildings for 2021)

In this study, we group the BSM and BSC typologies under an individual building typology, titled BSM/BSC, since using either mud mortar or cement mortar does not alter building flood resistance within the specific vulnerability models used in

this study (see Section 2.4). Using the same reasoning, we group RC-CCP and RC-WDS under one building typology labeled RC-CCP/WDS. We determine the proportions of the building typologies per municipality based on the 2011 census data (type of outer wall, type of foundation), as described in Mesta et al. (2022a). The exact height (number of stories) of each building is uncertain and is therefore randomly sampled using typology-specific empirical distributions or single values (not provided in Mesta et al., 2022a) that are derived from data collected for more than 20,000 buildings after the 2015 Gorkha Earthquake

by the NSET (2016). These distributions/values are defined as follows: two stories for A and W, between one and four stories (in the respective ratio 0.35:0.40:0.15:0.10) for brick buildings (i.e., BSM, BSC, RM), and between one and five stories (in the respective ratio 0.1:0.1:0.45:0.25:0.1) for concrete buildings (i.e., RC-CCP/WDS). We disaggregate the exposure data to match the 30 m spatial resolution of the urban map containing the 2021 built-up areas (see Figure 2).

### 3.2.2 Scenarios B, C, and D (population and buildings for 2031)

Future exposure for Scenario B is simulated using the proportions of different building typologies and the empirical distributions/values of building heights defined in Scenario A. We disaggregate the exposure data into the urban map containing the 2021 and 2031 built-up areas (see Figure 2).

Scenario C introduces two DRR measures that can reduce flood risk. The first DRR measure assumes that every existing building in a given floodplain is elevated by 0.2 m, a flood-risk mitigation measure proposed by Du et al. (2020) to reduce

flood losses in Shanghai, China. This elevation considers the construction of a 0.2m thick concrete platform above the ground floor, which constitutes a realistic (technically feasible) strategy to potentially reduce flood damage, especially in areas that experience low or moderate flood depths. Greater building elevations (as considered by Du et al., 2020) could prevent larger losses but would reduce household comfort (i.e., result in an excessive decrease in the floor-to-ceiling height). The second DRR measure consists of flood-hazard-informed land-use planning (i.e., the restriction of future urbanization in flood-prone

areas). All future buildings are distributed only to the 2031 built-up areas outside the corresponding 100-year and 1000-year floodplains. Flooding-informed future urbanization is a reasonable, feasible strategy, as the density of buildings in the selected (non-flooded) 2031 built-up areas does not exceed the density of buildings in the existing 2021 built-up areas.

Scenario D incorporates DRR measures that account for the multi-hazard-prone nature of Kathmandu Valley and can reduce both flood and seismic risk effectively. This means that it still includes the Scenario D structural retrofitting policies and building code enforcement seismic risk-mitigation interventions introduced in Mesta et al. (2022a) (i.e., A, BSM, BSC building typologies are replaced by RM; the RC-CCP typology is converted to RC-WDS), in addition to the flood-related DRR measures proposed in Scenario C.

### 3.2.3 Building replacement value

Building replacement value calculations in Mesta et al. (2022a) do not account for variable building heights. Based on the graphical description of buildings provided by Chaulagain et al. (2016) and the data reported by NSET (see Section 2.2.1), we assume that the values of area per building ($m^2$) included in Chaulagain et al. correspond to two-story buildings for A, W, BSM/BSC, and three-story buildings for RC-CCP/WDS and RC-WDS. This allows us to estimate a ratio of building area per story for each building, subsequently used to derive total costs for buildings with any number of stories. For simplicity, the building area and unit construction cost for the combined BSM/BSC typology (i.e., 75 $m^2$ and 250 €/$m^2$) are calculated as the average of the values for both individual typologies reported in Chaulagain et al. (2016). The building area and unit construction cost for the combined RC-CCP/WDS building typology (i.e., 82 $m^2$ and 318 €/$m^2$) are computed assuming a 76%-24% contribution of the values for both individual typologies reported in Chaulagain et al. (2016), in line with the relevant proportion of these typologies assumed in Mesta et al. (2022a). The additional costs associated with the elevation of buildings in Scenarios C and D are estimated based on the volume of concrete required to build the elevation platform (which is the platform thickness multiplied by the ratio of building area per story), considering a unit cost of 100 €/$m^3$ based on expert judgment.

**Table 1.** Summary of considered exposure scenarios for Kathmandu Valley

| Exposure scenario | Scenario A | Scenario B | Scenario C | Scenario D |
|---|---|---|---|---|
| Year | 2021 | 2031 | 2031 | 2031 |
| Population | 3,151,741 | 3,792,232 | 3,792,232 | 3,792,232 |
| Number of buildings | 789,898 | 943,606 | 943,606 | 943,606 |
| Aggregated replacement value (€) | 17,141,921,300 | 20,309,544,200 | 20,380,261,200* 20,409,943,100** | 26,442,366,000* 26,470,663,000** |
| Building typologies featured | A, W, BSM/BSC, RC-CCP/WDS | A, W, BSM/BSC, RC-CCP/WDS | A, W, BSM/BSC, RC-CCP/WDS | RC-WDS, RM |
| DRR actions | - | - | Elevating buildings, flood-hazard-informed land-use planning | Elevating buildings, flood-hazard-informed land-use planning, structural retrofitting, building code enforcement |

A: adobe; W: wood-frame; BSM: brick stone masonry with mud mortar; BSC: brick stone masonry with cement mortar; RM: reinforced masonry; RC-CCP: current-construction-practice reinforced concrete; RC.WDS: well-designed reinforced concrete.

Note: Scenarios C and D have two different aggregated replacement values, as the cost of elevating buildings in the 100-year (*) or the 1000-year floodplains (**) differ.

### 3.3 Modeling flood vulnerability

Since no specific flood vulnerability functions are developed for the study area, we adopt the depth-damage functions of the global flood depth-loss model developed by the Joint Research Center (JRC) of the European Commission (Huizinga et al., 2017). More sophisticated analytical flood fragility and vulnerability functions, which propagate uncertainties in the hazard-dependent failure of building components and associated repair/replacement costs (e.g., Nofal et al., 2020; Nofal and van de Lindt, 2021), would require detailed component-level vulnerability information that is not available for this study.

The JRC vulnerability functions were developed for distinct continents and building occupancies (e.g., residential, commercial, industrial). They express flood depth in meters and losses as mean loss ratios (i.e., financial losses as a percentage of the building replacement cost). We select the JRC vulnerability function for residential buildings in the Asian continent as our baseline function. We modify it to consider specific features of Kathmandu Valley's building stock. Firstly, we set a maximum damage to be 100% for A and W, and 60% for all brick (BSM/BSC, RM) and concrete (i.e., RC-CCP, RC- and WDS) typologies, following JRC recommendations. The 60% maximum damage threshold used for some typologies reflects the assumption that a flood cannot damage major water-resistant structural components, which represent a substantial portion of building construction costs (Huizinga et al., 2017). This assumption is in line with other studies such as the Central American

Probabilistic Risk Assessment (CAPRA) initiative (2012), which assigns 60% maximum flood losses to masonry and concrete buildings, and FEMA (2022), which indicates that major structural components are expected to withstand flood events. The damage thresholds are used to scale the loss values of the baseline function and derive two material-specific (i.e., non-resilient and resilient) vulnerability functions. Secondly, we adjust the two modified material-specific versions of the baseline function to account for different numbers of stories. Since JRC does not provide relevant information to infer the contribution of different building heights to losses, we assume that the baseline function is generally representative of two-story buildings and calculate appropriate height-adjustment factors in line with the procedure introduced in Gentile et al. (2022). These factors are used to multiply the loss values of the two material-specific functions and obtain the six vulnerability functions illustrated in Figure 5. The A building typology is highly vulnerable to material deterioration from prolonged contact with flood water (e.g., De Risi et al., 2013; Medero et al., 2011; Tiepolo and Galligari, 2021). The W building typology is also considerably vulnerable to floods (especially high-velocity flows; e.g., Becker Andrea B. et al., 2011). Both brick (BSM/BSC, RM) and concrete (i.e., RC-CCP, RC-WDS) typologies have higher durability compared with A and W, and low permeability and represent the most flood-resistant buildings (e.g., Balasbaneh et al., 2019; Li et al., 2016). Although the flood vulnerability may vary slightly between brick and concrete typologies (e.g., URM buildings are less able to resist the pressure of flood water exerted on walls than RM and RC buildings; Englhardt et al., 2019), these differences are not accounted for in the JRC vulnerability functions and thus are not included in this study.

The height-adjustment procedure of Gentile et al. (2022) assumes that the building replacement value is directly proportional to the number of stories, which may not be strictly valid (e.g., electrical and mechanical equipment are usually installed on the ground floor and mixed occupancy buildings can have commercial areas on the ground floor, increasing the relative replacement values of this story). Furthermore, the construction costs considered in this study exclude content costs, such that the resulting financial losses may be underestimated. However, it should be emphasized that the exact losses for a given exposure scenario (absolute or relative) are not strictly of interest in this study. Instead, we focus on producing comparable loss outputs for all exposure scenarios that are based on consistent assumptions. In this way, we aim to investigate how risk changes across the exposure scenarios in a relative sense. It is also worth noting that the uncertainty in the vulnerability model may strongly affect the loss estimation, particularly in terms of loss variability for a given mean return period. However, such uncertainty may be neglected if mean loss quantities are considered for comparison across different scenarios, as in this study.

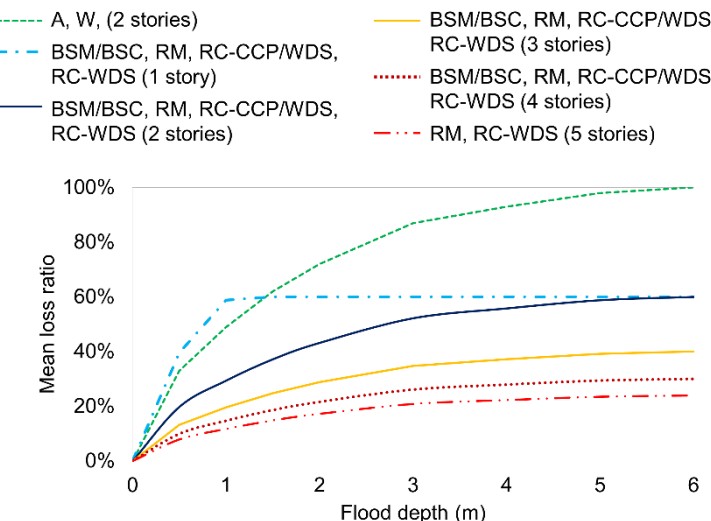

A: adobe; W: wood-frame; BSM: brick stone masonry with mud mortar; BSC: brick stone masonry with cement mortar; RM: reinforced masonry; RC-CCP: current-construction-practice reinforced concrete; RC-WDS: well-designed reinforced concrete.

**Figure 5.** Flood vulnerability functions for the different considered building typologies and their associated range of heights

Moreover, to avoid overestimating losses, we account for the difference between the ground level (above which flood depth is reported in the hazard maps) and the ground-floor level (above which flood depth is measured in the vulnerability functions). We set this difference at 0.2 m, as suggested in previous studies on flooding vulnerability for residential buildings (e.g., Dabbeek et al., 2020; Maqsood et al., 2014; Olsen et al., 2015) and after consulting construction blueprints of buildings in the study area. Furthermore, we use the procedure detailed in Mesta et al. (2022b) to classify populations per municipality as low, middle, or high income for facilitating socioeconomic disaggregation of financial losses. The classification is based on three variables (i.e., access to mobile/telephone services, mass media communication, and means of transportation) recorded in the 2011 Census, which are treated as proxies for economic wealth. The Census data are aggregated at the municipality level; therefore, any variability in the population's income level within each municipality is not (and cannot be) assessed. The classification is quantile, such that the three income categories contain an equal number of municipalities. We assume that the population's income level did not vary between 2011 and 2021 and would remain unchanged in 2031, given the lack of available data to make confident projections. This assumption is partially supported by previous work from Cutter and Finch (2008), who suggested that the social vulnerability of a community, which is influenced by its underlying socio-economic and demographic characteristics (e.g., income level, gender, age), is not expected to vary significantly over timeframes similar to those considered in this study.

## 4 Results

### 4.1 Distribution of buildings in the floodplain

Figure 6 (panel a) and Table 2 present a breakdown of the expected number of buildings and their proportions of the building stock within various depth ranges of the 100-year floodplain across the four exposure scenarios. In Scenario A, 108,922 buildings (14% of the 2021 building stock) are located in this floodplain, of which 80% and 20% respectively experience flood depths below and above 2.0 m. To provide some context, a 2.0 m flood depth produces mean loss ratios between 43% and 72% for non-elevated buildings with one or two stories and between 17% and 29% for non-elevated buildings with three, four, and five stories. In Scenario B, the larger building stock results in 130,106 buildings positioned within the floodplain (19% more than Scenario A), although the overall proportion of buildings in the floodplain (14% of the 2031 building stock) and their distribution by flood-depth range remains practically the same as in Scenario A. This means that, considering a 100-year flooding occurrence, future urban expansion (driven by the constraints of past planning decisions) is projected to continue occurring in both inundated and non-inundated areas of the valley. Scenarios A, C, and D yield identical results in Figure 6, since the flood-hazard-informed land-use planning imposed as part of Scenarios C and D means that the expected number of buildings within the floodplain in 2031 (Scenarios C, D) remains limited to 2021 levels (Scenario A). This measure decreases the proportion of flood-exposed buildings in both Scenarios C and D (grouped in Table 2) by 2.2%.

Figure 6 (panel b) and Table 2 present a breakdown of the expected number of buildings and their proportions of the building stock within various depth ranges of the 1000-year floodplain across the four exposure scenarios. In Scenario A, 117,179 buildings (15% of the 2021 building stock) are located in this floodplain, of which 66% and 34% respectively experience flood depths below and above 2.0 m. Scenario B results in 180,478 buildings positioned within the floodplain (54% more than Scenario A). In contrast with the 100-year flood-occurrence case, Scenario B shows a notable increase in the total proportion of buildings within the floodplain (19% of the 2031 building stock) relative to Scenario A. This means that, considering a severe flooding occurrence, future expansion (conditioned on past planning decisions) is projected to disproportionately occur in inundated areas. In Scenarios C and D, restricting future urban growth within the floodplain reduces the proportion of flood-exposed buildings by 6.7%.

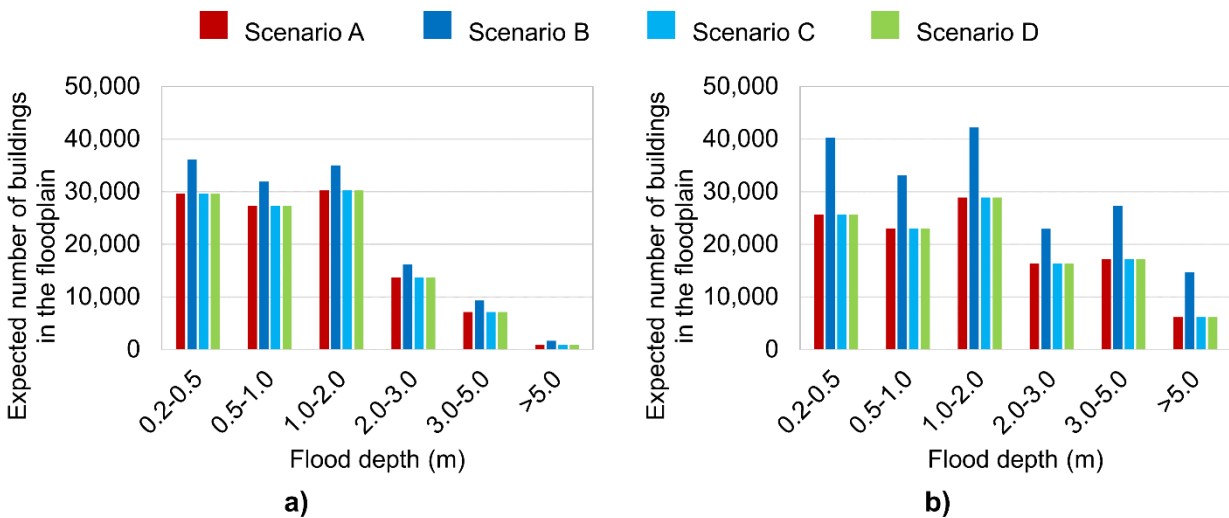

**Figure 6.** Exposure to flooding: the expected number of buildings within a given range of flood depth, per flooding occurrence and exposure scenario

**Table 2.** Exposure to flooding: proportions of the total building stock within a given range of flood depth, per flooding occurrence and exposure scenario.

| Flooding occurrence | Scenario | 0.2-0.5 m | 0.5-1.0 m | 1.0-2.0 m | 2.0-3.0 m | 3.0-5.0 m | >5-0 m |
|---|---|---|---|---|---|---|---|
| 100-year | A | 3.8% | 3.5% | 3.8% | 1.7% | 0.9% | 0.1% |
| | B | 3.8% | 3.4% | 3.7% | 1.7% | 1.0% | 0.2% |
| | C, D | 3.1% | 2.9% | 3.2% | 1.5% | 0.8% | 0.1% |
| 1000-year | A | 3.3% | 2.9% | 3.7% | 2.1% | 2.2% | 0.8% |
| | B | 4.3% | 3.5% | 4.5% | 2.4% | 2.9% | 1.6% |
| | C, D | 2.7% | 2.4% | 3.1% | 1.7% | 1.8% | 0.7% |

Figure 7 presents additional municipality-level insights on the spatial distribution of buildings within the 100-year floodplain. We identify a large variability in the percentage of buildings in inundated areas across the valley. For Scenario A, proportions of buildings within the floodplain are equal to or less than 10% for 56 municipalities, between 10% and 20% for 40 municipalities, and between 20% and 24% for 8 municipalities. Most municipalities with the largest number and proportions of buildings in inundated areas are located around the central and northern parts of the valley. Scenario B reflects the effects of not controlling future urbanization in flood-prone areas; proportions of buildings within the floodplain are equal to or less than 10% for only 44 municipalities, between 10% and 20% for 50 municipalities, and between 20% and 30% for 10 municipalities. In contrast, the proportions of buildings within the floodplain for Scenarios C and D are equal to or less than 10% for 79 municipalities, between 10% and 20% for 22 municipalities, and between 20% and 21% for 3 municipalities, reflecting the benefits of constraining future urbanization to non-inundated areas.

Figure 8 presents the municipality-level spatial distribution of buildings within the 1000-year floodplain. For Scenario A, proportions of buildings within the floodplain are equal to or less than 10% for 49 municipalities, between 10% and 20% for 43 municipalities, and between 20% and 28% for 12 municipalities. Corresponding Scenario B proportions are equal to or less than 10% for 31 municipalities, between 10% and 20% for 50 municipalities, and between 20% and 40% for 23 municipalities. Corresponding proportions for Scenarios C and D are equal to or less than 10% for 75 municipalities, between 10% and 20% for 24 municipalities, and between 20% and 25% for 5 municipalities.

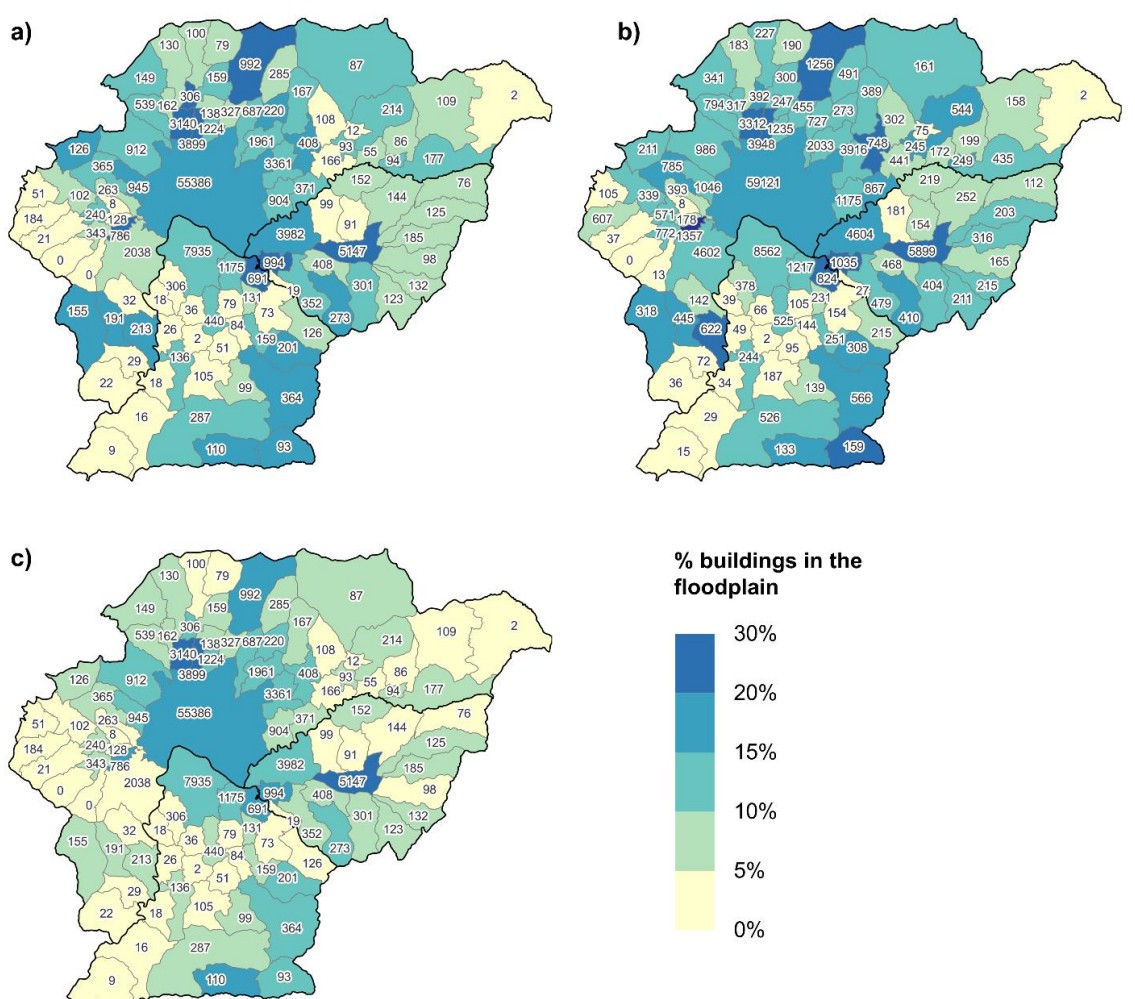

**Figure 7.** Spatial distribution of buildings in the 100-year floodplain for a) Scenario A, b) Scenario B, and c) Scenarios C and D. The numbers plotted inside each municipality correspond to the expected number of buildings in the floodplain.

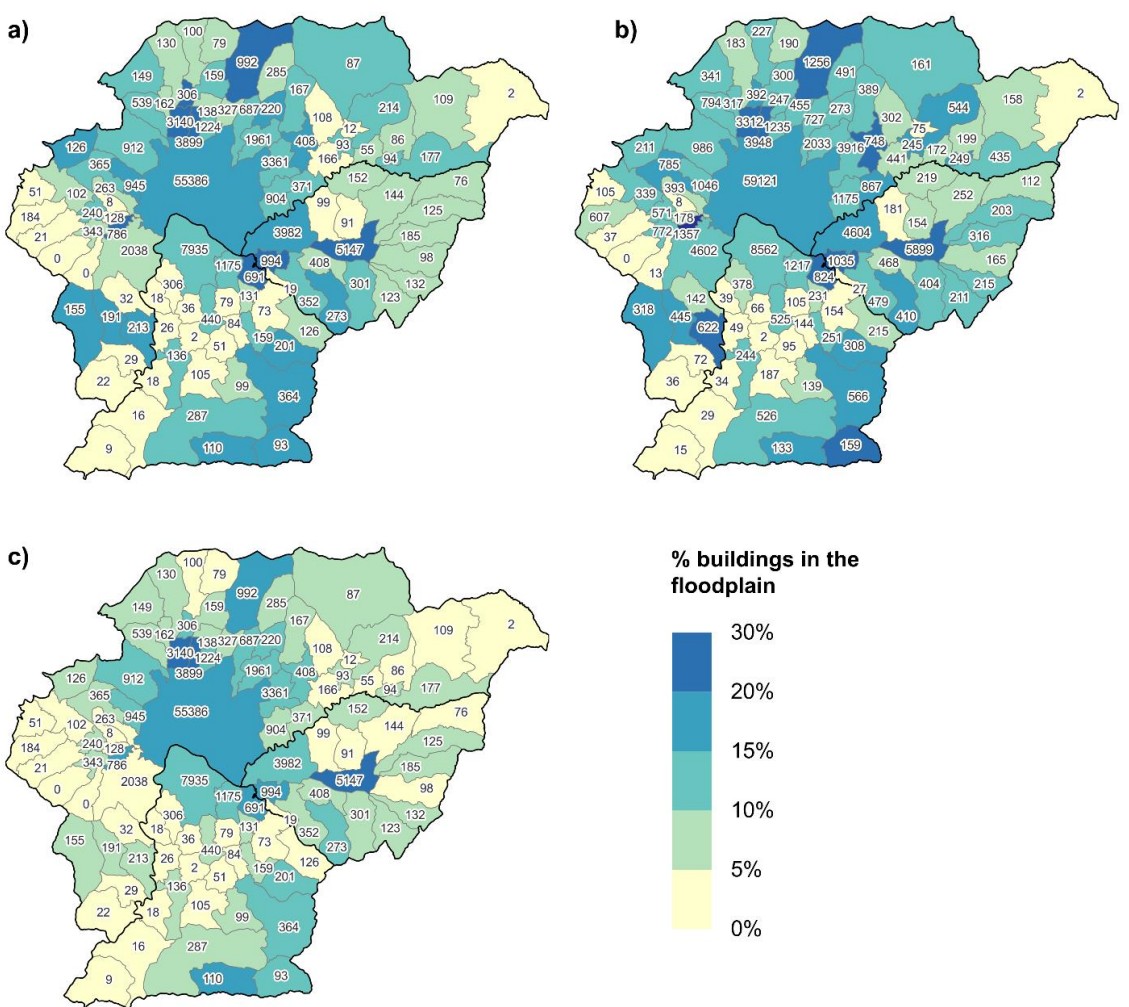

**Figure 8**. Spatial distribution of buildings in the 1000-year floodplain for a) Scenario A, b) Scenario B, and c) Scenarios C and D. The numbers plotted inside each municipality correspond to the expected number of buildings in the floodplain.

## 4.2 Losses

Figure 9 (panels a, b) presents the mean loss ratios associated with the 100-year flooding occurrence, disaggregated by district and income level. From panel a, we observe some variability in the mean loss ratios by district. The Bhaktapur district exhibits the largest mean loss ratios for all the scenarios, which is explained by its disproportionate share of exposure in inundated areas (there are only minor differences in the prevalence of different building typologies between districts, and the three districts are dominated by brick and concrete building typologies that are assumed to have the same level of flood vulnerability; see Section 2.3.) For instance, in Scenario A, the percentage of buildings in the floodplain is 14.9% in Bhaktapur, 14.5% in Kathmandu, and only 9.8% in Lalitpur. Proportions of buildings that experience flood depths below and above 2.0 m

respectively are 72%-28% in Bhaktapur, 84%-16% in Kathmandu, and 66%-34% in Lalitpur. Similar results are observed for Scenario B, because the overall proportion of buildings within different flood-depth ranges of the floodplain remains largely identical (see Table 2), and the building stock's vulnerability is not changed. While DRR measures implemented in Scenarios C and D reduce the mean loss ratios compared with those of Scenario B, the relative differences in mean loss ratios between districts are not particularly affected.

From Figure 9 (panel b), we identify some variability in the mean loss ratios by income level. All scenarios produce the highest mean loss ratios for the high-income population, which reflects their disproportionate share of buildings in inundated areas (there are also minor differences in the prevalence of building typologies between income groups, but the three income groups are dominated by brick and concrete typologies). For instance, in Scenario A, the proportion of buildings in the floodplain is 15% in high-income municipalities, 11% in middle-income municipalities, and 12% in low-income municipalities. Proportions of buildings that experience flood depths below and above 2.0 m, respectively, are 81%-19% for high-income municipalities, 75%-25% for middle-income municipalities, and 84%-16% for low-income municipalities. Scenario B shows similar results to Scenario A due to its similar proportions of buildings within different flood-depth ranges of the floodplain (see Table 2) and its identical quality of building stock. In addition, the benefits that result from the building elevation strategy and flood-hazard-informed land-use planning proposed in Scenario C are larger for the low-income population than for the other income groups; the mean loss ratio decreases from Scenario B to C by 44% for the low-income municipalities, by 28% for the middle-income municipalities, and by 22% for the high-income municipalities. There are two main reasons for this trend. On the one hand, low-income municipalities contain the largest proportion of flood-exposed buildings in areas with flood depths below 1.0 m, where the effects of the elevation strategy are more noticeable (as implied by the steep initial slopes of the vulnerability curves presented in Figure 5). In Scenario B, the proportion of buildings that experience flood depths below 1.0 m is 63% for low-income municipalities, 50% for middle-income municipalities, and 51% for high-income municipalities. On the other hand, the proportions of new Scenario B buildings located in the floodplain are higher across low-income municipalities (34% in total) than across middle-income (24%) and high-income (11%) municipalities. Furthermore, the benefits from the multi-hazard (i.e., flooding and seismic) risk-mitigation measures integrated within Scenario D are slightly better than those from the single-hazard-focused Scenario C: between Scenarios B and D, the mean loss ratio drops by 45% for the low-income municipalities, by 29% for the middle-income municipalities, and by 23% for the high-income municipalities.

Figure 9 (panels c, d) presents the mean loss ratios associated with the 1000-year flooding occurrence, disaggregated by district and income level. Similar to the 100-year flood case, there is an implicit relationship between the mean loss ratios and the extent of exposure in inundated regions; the same general trends for mean loss ratio across districts and income levels are observed for the more severe flooding occurrence. Bhaktapur district exhibits the highest mean loss ratios for all exposure scenarios, followed by Kathmandu and Lalitpur. All exposure scenarios result in the highest mean loss ratios for the high-income population, followed by the middle-income and low-income populations. The largest benefits from the risk mitigation strategies are also associated with the low-income population.

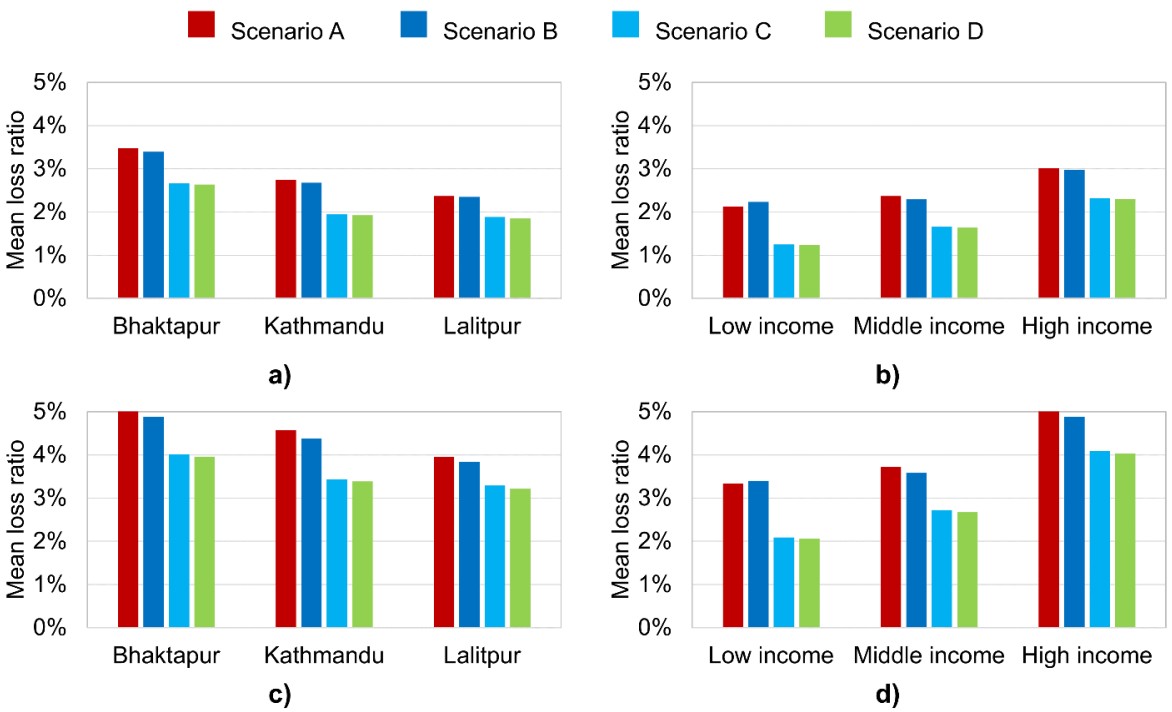

**Figure 9.** Mean loss ratios disaggregated by district (panels a, c) and income level (b, d), corresponding to the 100-year (upper panels) and 1000-year (lower panels) flooding occurrences

Table 3 summarizes the absolute changes in mean loss ratios and mean absolute financial losses (i.e., repair/reconstruction costs) for both flooding occurrences, considering Scenario A as a baseline. These results reveal how "no action" and implementing DRR measures could affect flood risk in Kathmandu Valley. In the 2021 exposure scenario, 3,151,741 people live in 789,898 buildings, with a total replacement value of € 17.1 billion. The 100-year mean absolute financial losses estimated for Scenario A are almost € 473 million (corresponding to a mean loss ratio of 2.8%), while the mean absolute

financial losses for the more severe 1000-year flooding occurrence are nearly € 775 million (corresponding to a mean loss ratio of 4.5%).

In the 2031 exposure scenarios, 3,792,232 people are allocated across 524 943,606 buildings, which have a total replacement value of € 20.3 billion in Scenario B, € 20.4 billion in Scenario C, and € 26.4 billion in Scenario D. Changes to mean absolute financial losses associated with the 100-year flooding occurrence and the 2031 exposure scenarios are as follows: they increase

by more than € 74 million (+16%) in Scenario B, decrease by more than € 63 million (-13%) in Scenario C, and rise by more than €52 million in Scenario D (+11%), relative to Scenario A. For the 1000-year flooding occurrence, mean absolute losses increase by nearly € 108 million (+14%) in Scenario B, decrease by more than € 66 million (-9%) in Scenario C, and rise by more than € 130 million in Scenario D (+17%), relative to Scenario A. The relative increase in mean absolute financial losses for Scenario B is due to the presence of more assets in the floodplain. In other words, Scenario B demonstrates that a larger

population can easily lead to greater flood losses when risk mitigation is neglected. In contrast, the relative decrease in mean absolute financial losses for Scenario C shows that, despite a growing population, elevating existing buildings and implementing flood-hazard-informed land-use planning could significantly reduce flood losses in the future. However, it should be noted that risk-mitigation actions implemented in Scenario C would still leave the building stock highly vulnerable to earthquakes, and thus do not completely address multi-hazard risk in the valley, which is left to Scenario D. Note that a previous study by the authors (Mesta et al., 2022a) revealed that not implementing seismic risk-mitigation actions for the valley (i.e., analogous to Scenario C in this study) could increase mean absolute financial seismic losses in the future (2031) by more than € 1.7 billion (+20%) relative to equivalent current levels. In contrast, improving the seismic strength of buildings (i.e., similar to Scenario D in this study), could reduce mean absolute financial seismic losses in the future by more than € 1.1 billion (-14%) relative to equivalent current levels. The relative increase in mean absolute financial losses in Scenario D is associated with the larger replacement value of its building stock (due to the structural retrofitting and building code enforcement measures implemented). This highlights a tension between short-term up-front costs (incurred before the occurrence of hazard events) and long-term benefits (after the occurrence of hazard events) associated with holistic DRR measures. In summary, Scenario D demonstrates that, despite a growing population, adequate DRR measures that aim to improve the building stock's quality (for better sustaining both flood and earthquake damage) as well as incentivize urbanization away from flood-sensitive areas can limit (but not reduce) mean absolute financial flood losses in the future.

Absolute changes to the mean loss ratios provide additional interesting findings. In Scenario A, the mean loss ratios associated with the 100-year and 1000-year flooding occurrences are 2.8% and 4.5%, respectively. In Scenario B, as future urbanization continues occurring in both inundated and non-inundated areas and there are no changes in the building stock's quality, the mean loss ratios only show minimum variations compared to Scenario A. In Scenario C, elevating buildings and the promotion of flood-hazard-informed land use produce a significant decrease in the mean loss ratios, which drop to 2.01% and 3.5% (27% and 23% smaller than in Scenario A), respectively. Due to additional improvements in the building stock's quality in Scenario D, the mean loss ratios drop further to 1.99% and 3.4% (28% and 24% smaller than in Scenario A, respectively). By comparing the mean loss ratios from both Scenarios C and D relative to Scenario A, we notice that seismic-risk mitigation interventions by themselves do not contribute much to reducing flood risk in the valley due to the low replacement rate (<5%) of non-flood-resilient buildings (i.e., A, W) with flood-resilient buildings (i.e., RM) as a result of the seismic upgrading process. However, it is important to remember that Scenario D represents a much more robust approach to multi-hazard risk mitigation than Scenario C.

**Table 3.** Mean loss metrics for Scenario A, and absolute changes to these metrics in Scenarios B, C, and D

| Flooding occurrence | Metric | Scenario A | Scenario B | Scenario C | Scenario D |
|---|---|---|---|---|---|
| 100-year | Mean absolute financial losses (€) | 472,932,965 | + 74,654,044 | -63,283,799 | + 52,691,045 |
| | Mean loss ratio | 2.8% | -0.06% | -0.75% | -0.77% |
| 1000-year | Mean absolute financial losses (€) | 774,793,163 | + 107,901,808 | -66,393,767 | + 130,500,162 |
| | Mean loss ratio | 4.5% | -0.17% | -1.0% | -1.1% |

Figure 10 and Figure 11 present additional insights on the municipality-level spatial distribution of the mean loss ratio for the 100-year and 1000-year flooding occurrences, respectively. The alignment between the proportion of buildings located in inundated areas and the mean loss ratios is clear when the maps from Figure 10 and Figure 11 are compared with those from Figure 7 and Figure 8; many municipalities with the largest mean loss ratios are situated in or around the central part of the valley. However, municipalities with notable mean loss ratios are not always associated with the largest proportions of buildings in inundated areas; some are simply subjected to relatively high flood depths (see Figure S1 and Figure S2 in the Supplementary Material).

Figure 12 illustrates the absolute changes to the municipality-level mean loss ratios for the 100-year flooding occurrence, considering Scenario A as a baseline. In Scenario B, the mean loss ratios show small absolute variations (between -1.0% and +1.2%) compared to Scenario A, since future urbanization continues occurring in both flooded and non-flooded areas. Some municipalities experience a decrease in mean loss ratio (see Figure 7), where future urbanization outside the floodplain is larger than that within it. The relative effects of the building elevation strategy and the flood-hazard-informed land-use planning proposed in Scenario C are noticeable: absolute reductions in mean loss ratios for Scenario C relative to Scenario A ranges between 2.0-2.9% in five municipalities, between 1.0-2.0% in 18 municipalities, and are less than 1.0% in the remaining 81 municipalities. The benefits of implementing additional multi-hazard DRR measures in Scenario D are almost equivalent to those in Scenario C because the seismic upgrading of the building stock does not contribute much to reducing flood risk. Figure 13 presents the absolute changes to the municipality-level mean loss ratios for the 1000-year flooding occurrence considering Scenario A as a baseline. In Scenario B, the mean loss ratios exhibit some absolute variations (between -1.0% and +2.3%) relative to Scenario A, which are larger than in the 100-year flood case; in other words, the consequences of not controlling future urbanization in flood-prone areas can increase with the severity of the considered flooding occurrence. The effects of the flood-specific DRR measures implemented in Scenario C are as follows: absolute reductions in mean loss ratios for Scenario C relative to Scenario A ranges between 2.0-5.6% in 9 municipalities, between 1.0-2.0% in 32 municipalities, and are less than 1.0% in the remaining 63 municipalities. The benefits of the combined DRR measures in Scenario D are comparable to those in Scenario C.

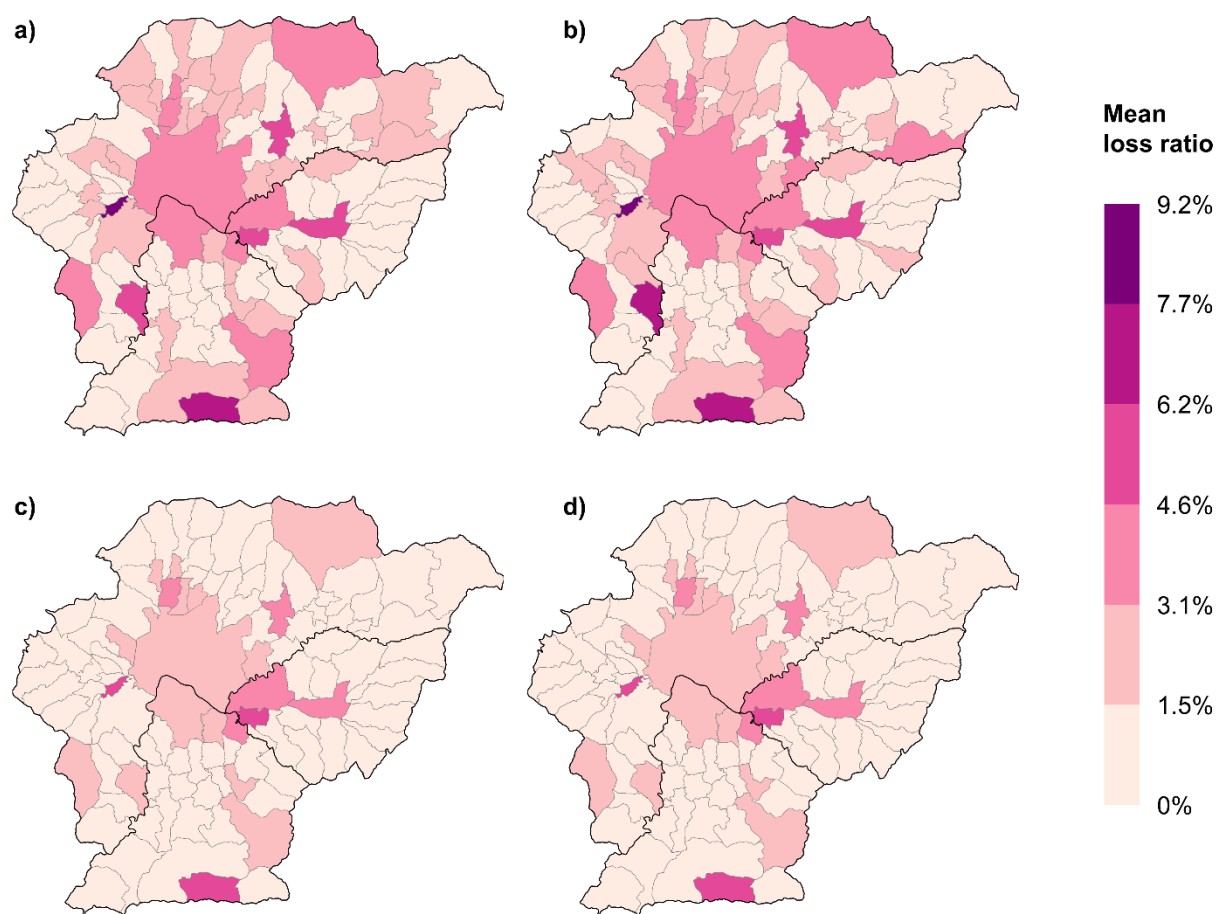

**Figure 10.** Spatial distribution of mean loss ratios, associated with the 100-year flooding occurrence, for a) Scenario A; b) Scenario B; c) Scenario C; and d) Scenario D

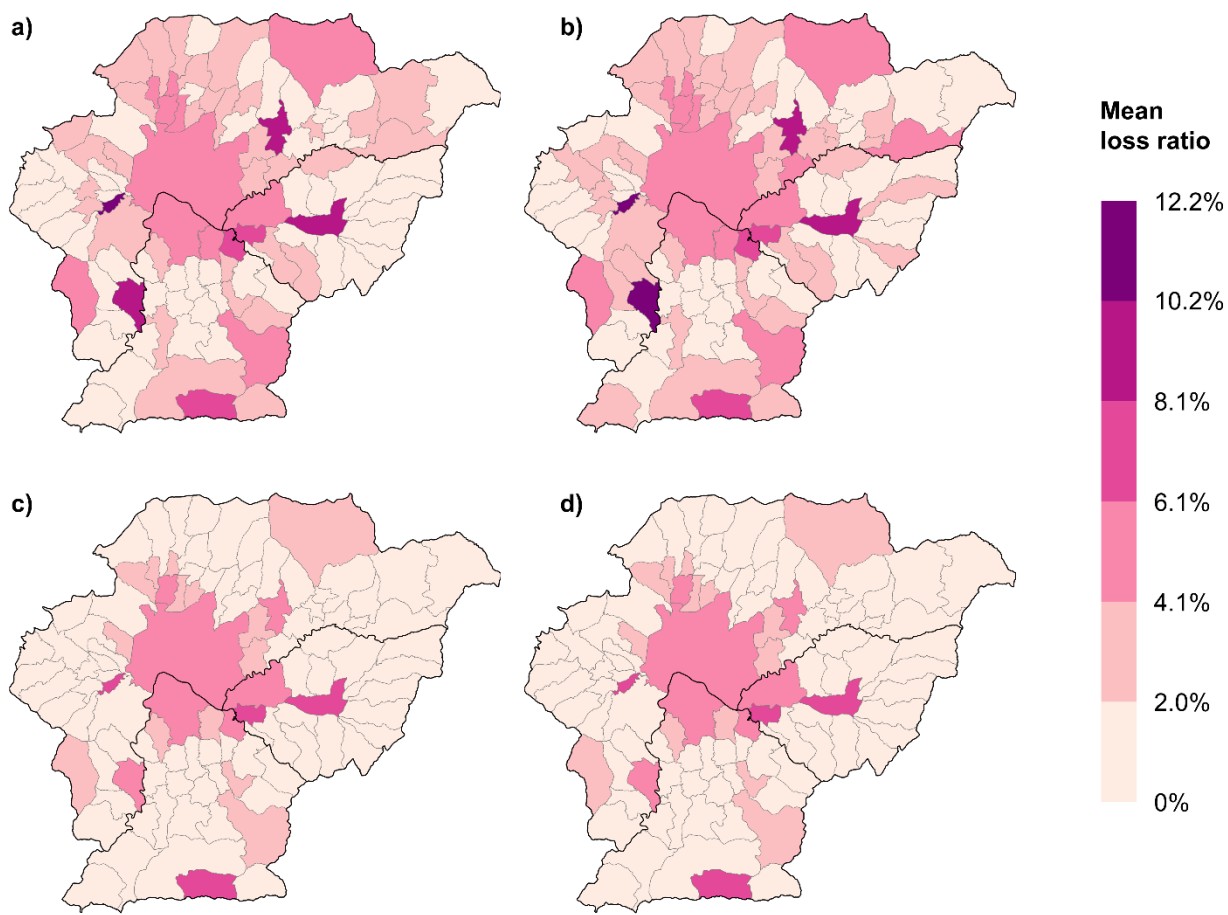

**Figure 11.** Spatial distribution of mean loss ratios, associated with the 1000-year flooding occurrence, for a) Scenario A; b) Scenario B; c) Scenario C; and d) Scenario D

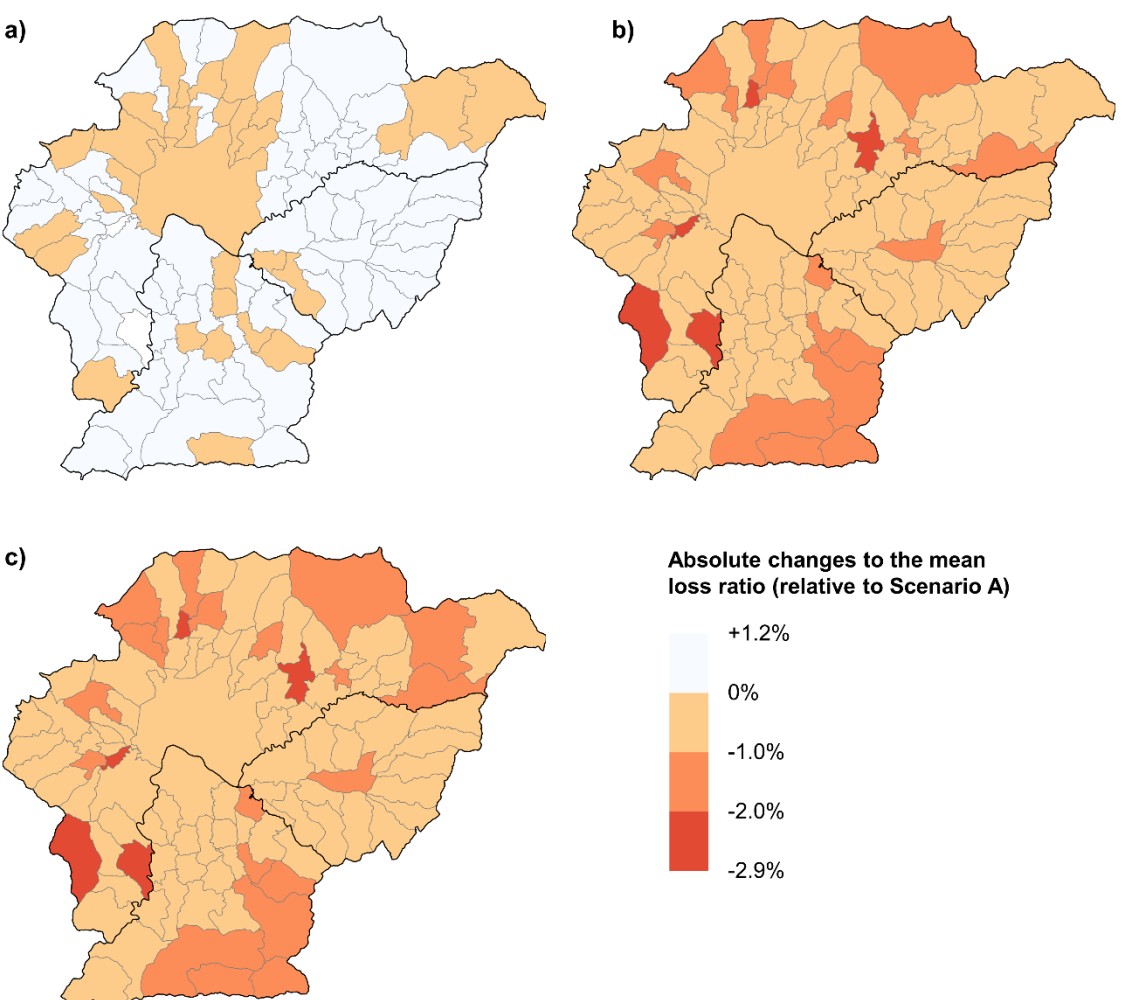

**Figure 12.** Absolute changes to the municipality-level mean loss ratios for a) Scenario B; b) Scenario C; c) Scenario D, relative to Scenario A, for the 100-year flooding occurrence

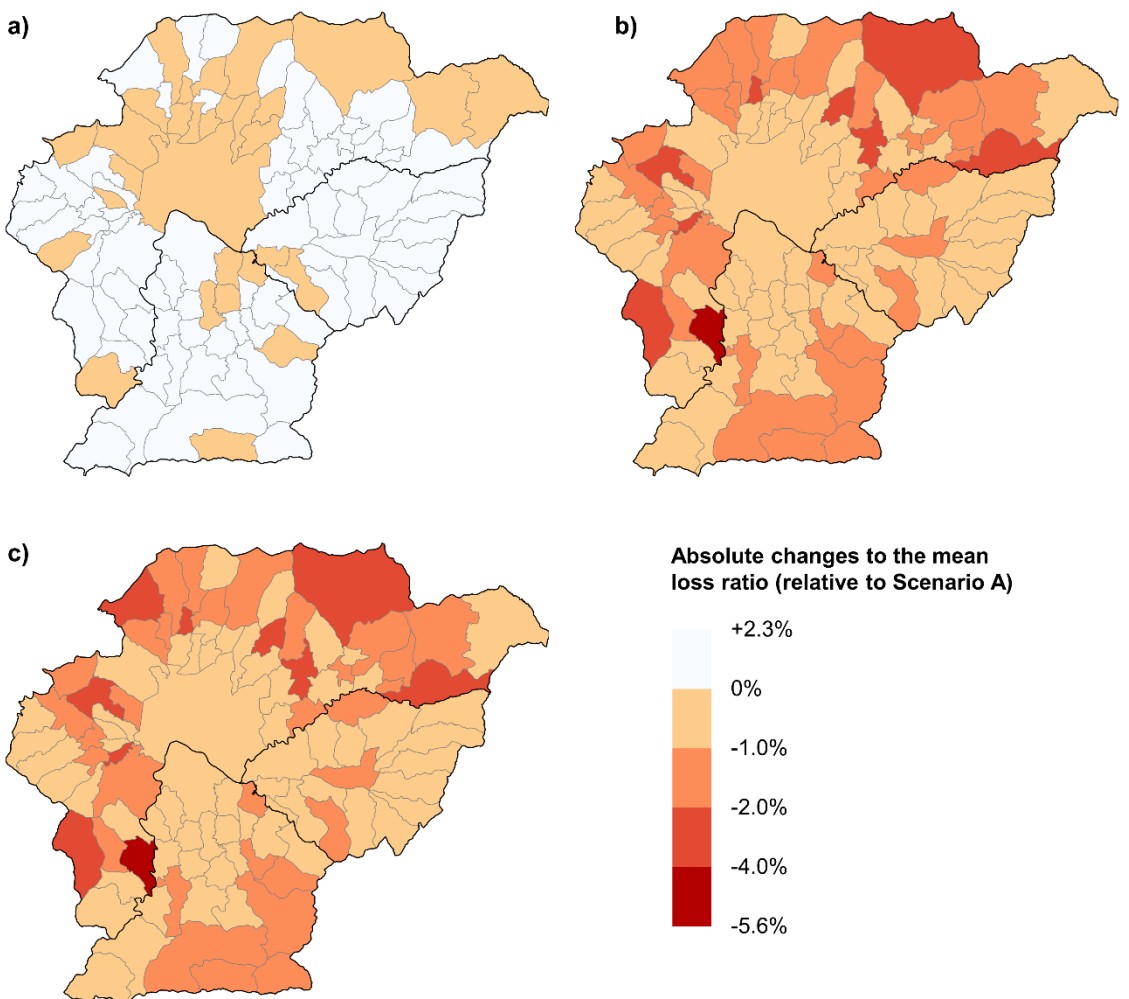

**Figure 13.** Absolute changes to the municipality-level mean loss ratios for a) Scenario B; b) Scenario C; c) Scenario D, relative to Scenario A, for the 1000-year flooding occurrence,

## 5 Discussion

The main results of this study provide a clear description of the current and potential near-future flood risk in Kathmandu Valley, suggesting that decision-makers of today have a unique opportunity to positively influence the risk of tomorrow, through their choices on implementing policies that control future risk drivers (e.g., Cremen et al., 2022b). However, we acknowledge that different sources of uncertainty and limitations of the data and methods used can influence the accuracy of the results obtained.

In this study, we characterize the flood hazard using global maps with a coarse resolution (i.e., 90 m), which may not capture the highly localized nature of flood hazard (e.g., associated with small streams). While finer resolution hazard maps (e.g., 10 m or lower) are generally preferred for conducting regional flood risk assessments, the spatial resolution of the hazard model must also be consistent with the resolution of the exposure model used. We characterize exposure in the valley using urban maps with a spatial resolution of 30 m; therefore, our analyses would not benefit from hazard maps of a finer resolution. In addition, some authors (e.g., Fatdillah et al., 2022; Zhang, 2020) report that using finer-resolution digital elevation models (DEM), which would be needed to produce finer-resolution flood hazard maps, can result in larger simulated flooded areas and losses compared to coarser-resolution DEM; however, other authors (e.g., McClean et al., 2020) suggest the opposite, indicating that flood risk may be exaggerated using flood maps based on global coarse DEM. These ambivalent findings suggest that the advantages of using finer-resolution flood maps for regional flood risk assessments, in fact, require careful evaluation for each specific context. Another limitation of the flood maps employed in this study is that they do not capture the effects of urbanization on flood hazard (i.e., the replacement of natural ground with impermeable surfaces, changes to drainage or irrigation systems, and deforestation can increase runoff during precipitation events). The use of physics-based flood simulations that include future urban footprints would address this issue (e.g., Jenkins et al., 2022) but may entail a significant computational cost.

Uncertainties and limitations associated with the exposure and vulnerability models also affect the loss outputs. Due to the absence of a reliable database for Kathmandu Valley containing exact building footprints (and relevant attributes such as building typology and height), we construct our exposure model by downscaling data collected from census and surveys (mainly at the municipality level) into the built-up areas of the valley (i.e., dasymetric mapping). While exposure disaggregation techniques are widely used in regional risk assessments (e.g., Geiß et al., 2022; Dabbeek et al., 2020), it is recommended to use original exposure models that are refined from the outset, since the accuracy of damage and loss estimates are highly sensitive to that of the exposure data. The accuracy of the exposure model may be of particular importance for flood loss assessments, given the potentially significant localized variability of flood hazard (i.e., flood depths can abruptly change even between closely-spaced locations). Moreover, the loss accuracy strongly depends on the quality of the vulnerability curves. In this study, we modify existing continental-based vulnerability curves to include relevant characteristics of the local building stock in Kathmandu Valley (e.g., building typology and height). However, it is difficult to ascertain how much (if any) uncertainty and/or accuracy is effectively improved with these modifications.

The design and implementation of risk-mitigation strategies also face several challenges. For instance, policies that restrict future urbanization within floodplains rely on the accuracy of spatial designations made within flood maps. While flood maps provide a good basis for floodplain management, regulation, and mitigation- e.g., in the USA, 100-year flood maps are used to identify Special Flood Hazard Areas where the National Flood Insurance Program's floodplain management regulations must be enforced (Ludy and Kondolf, 2012; Federal Emergency Management Agency (FEMA), 2010))- it is essential to acknowledge that different sources of uncertainty (e.g., climate change impacts, uncertainty in the hydrological/hydraulic

models, etc.) can affect the resulting floodplain delineation (Zahmatkesh et al., 2021). Consequently, populations outside the designated floodplains may still be at risk of flooding and should be made aware of this.

A comprehensive sensitivity analysis could be conducted to investigate the impact of the aforementioned limitations on the results (e.g., Bernhofen et al., 2022). However, since the main focus of this study is to investigate relative risk changes across different sets of DRR-related actions, the uncertainty associated with the absolute losses is not within the scope of this study.

The results obtained in this study provide valuable information for decision makers about drivers of exacerbated future flood risk and can help to support appropriate policy making. The proposed framework could also inform high-level guidelines for identifying flood risk hotspots that deserve a more detailed local DRR assessment (e.g., including higher-resolution data/models, a proper analysis of costs, a tailored analysis of DRR measures, etc.).

## 6 Conclusions

This study has examined the present (2021) and future (2031) flood risk in Kathmandu Valley, considering 100-year and 1000-year mean return period flooding occurrences. Different assumptions on the estimated population, number of households, and building stock quality have been made to construct four plausible current and near-future urban development states for the valley.

The key findings of this study are as follows. First, results reveal that a significant proportion of the current building stock is located within the 100-year and 1000-year floodplains (14% and 15%, respectively), which may lead to significant losses. However, an appropriate combination of DRR measures (i.e., building elevation and flood-hazard-informed land-use planning) can substantially limit mean absolute financial losses and reduce relative versions of these losses (i.e., expressed as a proportion of associated replacement costs) in the future, compared to equivalent current levels. Second, this study reveals that high-income populations are exposed to the highest mean loss ratios across both flooding-occurrence cases due to having the largest proportions of buildings in the floodplain. This contrasts with the trend in income versus earthquake-related losses identified for the same region in previous work (Mesta et al., 2022a), where low-income populations exhibited the highest seismic risk. This discrepancy illustrates that risk-mitigation measures can have varying effects for different hazards; therefore, DRR plans should be appropriately tailored for a specific region or sub-region and account for multiple hazards. Kathmandu Valley's building stock is highly vulnerable to earthquakes due to the prevalence of URM buildings (particularly in low-income municipalities), such as adobe and brick/stone masonry. However, this feature of the building stock does not make it particularly susceptible to flood damage (except in the case of adobe houses, which are made of mud), which is why a multi-hazard approach to DRR that also considers earthquake vulnerability strengthening measures has little effect on the mean loss ratios (and even results in increased mean absolute financial losses) in this study. Instead, the flood risk is mainly controlled by the extent to which populations are located in the floodplain. Considering that hazard intensities vary spatially and that flooding and earthquake-induced ground shaking can affect different proportions of buildings in a given municipality,

combinations of individual DRR measures should be investigated to find the optimal DRR solution for a given municipality. Third, this study demonstrates that DRR initiatives uniformly targeting flood risk across different income levels produce the largest benefits for low-income populations. These findings are relevant because the benefits of mitigation measures are currently not well understood/quantified by various stakeholders in Nepal In summary, this work provides important insights for decision-makers on how effective risk-informed policy making can limit future flood risk compared to current levels,
particularly for low-income populations.

While this paper is focused on two levels of flooding occurrence, future research could analyze further scenarios to provide more robust results. Nonetheless, we do not expect the general trends identified in this study to significantly differ for other flood occurrence cases. Fine-resolution local hazard models (if and when available) could be used to more accurately quantify flood hazard (and the associated risk), explicitly including the effect of building footprints, climate change, etc. Moreover,
updated census information (when available) could be employed to adjust present and future exposure estimations. In addition, the accuracy of the characterization of physical vulnerability could be improved through appropriate modifications to the selected vulnerability functions in line with local construction practices. Future research could also investigate the effectiveness of other possible flood-related DRR actions (e.g., ring dike, wet-proofing, dry-proofing, nature-based solutions, relocation). Lastly, while the benefits of risk-mitigation plans have been discussed in this paper without a proper analysis of costs, various
methods such as cost-benefit evaluations (e.g., de Ruig et al., 2019; Du et al., 2020; Gentile and Galasso, 2021; Lasage et al., 2014; Scussolini et al., 2017) or multi-criteria decision making (e.g., Ahmadisharaf et al., 2016; Cremen et al., 2022a; Ruangpan et al., 2021), can help in selecting optimal risk-mitigation solutions.

In summary, this paper addresses the essential need to communicate the growing flood risk in Kathmandu Valley and potentially encourage local (or even Nepal-wide) risk-mitigation efforts. The adopted methodology can be easily extended to
other geographical contexts to quantify the impacts of other (multiple) natural hazards on the present and future built environment, providing decision makers with an adequate understanding of the risk consequences of particular actions and the importance of particular risk mitigation/adaptation strategies (Cremen et al., 2022b; Galasso et al., 2021).

*Data availability.* The flood hazard maps are available online through the METEOR project (available at https://maps.meteor-
project.org/map/flood-npl/, last accessed December 2022). The urban maps for Kathmandu Valley are available online through a public repository (available at https://doi.org/10.5281/zenodo.7406981, last accessed December 2022). OpenStreetMap (OSM) data are distributed under the Open Database License (OdbL) (openstreetmap.org/copyright). Other datasets used and/or analyzed during the current study are available from the corresponding author on reasonable request.

*Author contributions.* C.M., G.C., and C.G. conceived and designed the research. C.M. drafted the written content of the manuscript, performed the calculations, and developed the figures. All authors reviewed the manuscript.

*Competing interests.* The authors declare no competing interests.

*Acknowledgments.* We would like to thank Andrew Smith and Joseph Paul from Fathom for providing the flood hazard data. Carlos Mesta was supported by a Research Scholarship from the European Centre for Training and Research in Earthquake Engineering (EUCENTRE). Carmine Galasso and Gemma Cremen acknowledge funding from UKRI GCRF under grant NE/S009000/1, Tomorrow's Cities Hub.

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
