# Peer review of "Quantifying the potential benefits of risk-mitigation strategies on future flood losses in Kathmandu Valley, Nepal"

_EGUsphere, 2022_

## Author Comment (AC1)

**REVIEWERS' COMMENTS TO THE AUTHOR:**

**Reviewer 1:**

1.  **General comment**. **Mesta et al. construct current and near-future urban development states (in total four exposure scenarios) for the Kathmandu Valley and assess the flood risk using flood inundation maps of the 100-year and 1000-year return level (pluvial and fluvial combined). In my opinion, the study has the potential to become a valuable contribution to risk research in the area. However, there are several points that need clarification and improvement before it can be considered for publication.**

*Many thanks for your overall positive assessment of our manuscript and your insightful comments, which are addressed in detail below.*

**Major comments**.

2.  **Comment 1: Unspecific key result**. **The goal of the study (as far as I understood) is to provide decision makers with an adequate understanding of the risk consequences of particular actions. However, to me it was not clear what actually your key findings are. What is the new information that your study provides? What can decision-makers learn from your study? What is your key message to them? I image something like 5 bullet points summarizing the key findings of work**.

*Thank you for pointing this out. Indeed, the study's primary goal (and our key message to readers, including decision-makers) is to demonstrate how risk-mitigation measures can substantially limit/reduce flood-induced losses in the future, compared to equivalent current levels. Please note that this and other key findings of our work were already presented in the second paragraph of the Conclusions (L533-L555), but we have slightly modified this paragraph to better emphasize our message. The paragraph now reads as follows, with modifications marked in bold:*

*"**The key findings of this study are as follows. First,** results reveal that a significant proportion of the current building stock is located within the 100-year and 1000-year floodplains (14% and 15%, respectively), which may lead to substantial losses. However, an appropriate combination of DRR measures (i.e., building elevation and flood-hazard-informed land-use planning) can substantially limit mean absolute financial losses and reduce relative versions of these losses (i.e., expressed as a proportion of associated replacement costs) in the future, compared to equivalent current levels. **Second,** this study reveals that high-income populations are exposed to the highest mean loss ratios across both flooding-occurrence cases due to having the largest proportions of buildings in the floodplain. The trend in income versus flood-related losses contrasts with the trend in income versus earthquake-related losses identified for the same region in previous work (Mesta et al., 2022a), where low-income populations exhibited the highest seismic risk. This discrepancy illustrates that risk-mitigation measures can have varying effects for different hazards; therefore, DRR plans should be appropriately tailored for a specific region or sub-region and holistically account for multiple hazards. Kathmandu Valley's building stock is highly vulnerable to earthquakes due to the prevalence of URM buildings (particularly in low-income municipalities), such as adobe and brick/stone masonry. However, this feature of the building stock does not make it particularly susceptible to flood damage (except in the case of adobe houses, which are made of mud). This underlines why a multi-hazard approach to DRR that also considers earthquake vulnerability strengthening measures has little effect on the mean loss ratios (and even results in increased mean absolute financial losses) in this study. Instead, the flood risk is primarily controlled by the extent to which populations are located in the floodplain. Considering that hazard intensities vary spatially and that flooding and earthquake-induced ground shaking can affect different proportions of buildings in a given municipality, combinations of individual DRR measures should be investigated to find the optimal DRR solution for a given municipality. **Third,** this study*

*demonstrates that DRR initiatives uniformly targeting flood risk across different income levels produce the largest benefits for low-income populations. These findings are relevant because the benefits of mitigation measures are currently not well understood/quantified by various stakeholders in Nepal.* **In summary, this work provides important insights for decision-makers on how effective risk-informed policymaking can limit future flood risk compared to current levels, particularly for low-income populations.**

3. **Comment 2: Lack of discussion. I general, I miss a bit a critical discussion of the data and methods used. Some aspects need to be discussed in more detail. Justify better why usage of global and low resolution data sets for regional risk assessment (Line 115). Discuss limitations of flood maps. You state yourself that resolutions of 10 m or finer are recommended. Also you state that 'urbanization effects on flood hazard' are neglected (Line124). Please discuss influence on your results.**

*Thank you for your this comment. As we state in section "2.2 Hazard Modelling"* *(L137-L139), "the primary purpose of this study is to test different exposure/vulnerability scenarios using a common flood hazard input that is open and easily accessible; developing a bespoke fine-resolution hazard model for the study area is not within the scope of this work".* *Thus, the need for a detailed justification of the hazard model used is not warranted in this context, in our opinion.*

*However, based on this suggestion (and that of another reviewer), we have added a new section "4. Discussion" (between the "Results" and "Conclusions" sections), where we comment on the main limitations of this study (including all of those mentioned by the reviewer in this comment). Section "4. Discussion" (L483-L527) reads as follows:*

*"**4 Discussion***

[revised manuscript text omitted]

*References:*

1. *Bernhofen, M.; Cooper, S.; Trigg, M.; Mdee, A.; Carr, A.; Bhave, A.; Solano-Correa, Y.; Pencue-Fierro, E.; Teferi, E.; Haile, A.; Yusop, Z.; Alias, N.; Sa'adi, Z.; Bin Ramzan, M.; Dhanya, C.; Shukla, P. The Role of Global Data Sets for Riverine Flood Risk Management at National Scales. Water Resources Research, 2022.*
2. *Cremen, G., Galasso, C., and McCloskey, J.: Modelling and Quantifying Tomorrow's Risks from Natural Hazards, Sci. Total Environ., 2022b.*
3. *Dabbeek, J., Silva, V., Galasso, C., and Smith, A.: Probabilistic earthquake and flood loss assessment in the Middle East, Int. J. Disaster Risk Reduct., 49, 101662, 2020.*
4. *Fatdillah, E., Rehan, B., Rameshwaran, P., Bell, V., Zulkafli, Z., Yusuf, B., Sayers, P. Estimates of Flood Damage and Risk Are Influenced by the Underpinning DEM Resolution: A Case Study in Kuala Lumpur, Malaysia. Water, 14, 2208, 2022.*
5. *Federal Emergency Management Agency (FEMA): Special Flood Hazard Area (SFHA). Available at https://www.fema.gov/glossary/special-flood-hazard-area-sfha, last access: 27 November 2022.*
6. *Geiß, C., Priesmeier, P., Aravena Pelizari, P., Soto Calderon, A. R., Schoepfer, E., Riedlinger, T., Villar Vega, M., Santa María, H., Gómez Zapata, J. C., Pittore, M., So, E., Fekete, A., and Taubenböck, H.: Benefits of global earth observation missions for disaggregation of exposure data and earthquake loss modeling: evidence from Santiago de Chile, Nat. Hazards, 2022.*
7. *Jenkins, L; Creed, M., Tarbali, K., Muthusamy, M.; Šakić, R.; Phillips, J.; Watson, S.; Sinclair, H.; Galasso, C.; McCloskey, J. Physics-based simulations of multiple natural hazards for risk-sensitive planning and decision making in expanding urban regions. International Journal of Disaster Risk Reduction, 103338, 2022.*
8. *Ludy, K, Kondolf, G.M.: Flood risk perception in lands "protected" by 100-year levees. Nat. Hazards 61, 829–842, 2012.*

9. *McClean, F., Dawson, R., & Kilsby, C. Implications of using global digital elevation models for flood risk analysis in cities. Water Resources Research, 56, e2020WR028241, 2020.*
10. *Zahmatkesh Z., Han, S., Coulibaly, P.: Understanding Uncertainty in Probabilistic Floodplain Mapping in the Time of Climate Change. Water 13, 1248, 2021.*
11. *Zhang, Y. Using LiDAR-DEM based rapid flood inundation modelling framework to map floodplain inundation extent and depth. Journal of Geographical Sciences, 30, 1649-1663, 2021.*

4. **Comment 2: Lack of discussion. Futhermore, I miss a discussion on the merging of fluvial and pluvial floods maps. Can you just do that and assume that the only thing that matters is the water level? What about velocity? I would find it very interesting to learn more about the importance of those two types that you merged. How much of the flooded area is from pluvial floods? Are there a lot of areas that are exposed to both?**

*Thank you for this comment. We have added some lines (**in bold**) (L146-L164) to the second paragraph of section "2.2 Hazard Modelling", which clarify that the merging of fluvial and pluvial flood maps into aggregated hazard maps is based on a previous study (Tate et al., 2021). Note that we aggregate two independent events (fluvial, pluvial), which is acceptable from a statistical standpoint. In addition, we have provided a brief description of the aggregated hazard maps and noted that the effect of pluvial flooding largely dominates the aggregated flood maps. Also, we have added some references there that justify our use of flood depth as the sole intensity measure; flood depth is the most widely used intensity measure in flood loss estimation, for instance.*

*"(…) We combine individual flood maps into aggregated hazard maps that represent fluvial-pluvial flooding for each mean return period by taking their maximum depths in line with the method of Tate et al. (2021), **who mosaiced fluvial and pluvial flood grids to generate an aggregated flood hazard map for the United States.** The fluvial-pluvial hazard maps for each considered mean return period are presented in Figure 2. Hereafter, we describe the flooding-occurrence cases using only the terms "100-year" and "1000-year", omitting the description "mean return period" for brevity. **Overall, the aggregated flood maps are largely dominated by the effects of pluvial flooding: in both 100-year and 1000-year aggregated flood maps, around 15% of the flooded areas are exposed to both types of flooding, 84% are only exposed to pluvial flooding, and less than 1% are only exposed to fluvial flooding. It should be noted that fluvial flooding generally results in low-velocity flows dominated by hydrostatic pressure, while pluvial flooding often features higher flow velocities (Gentile et al., 2022); these differences in velocity characteristics could be important for estimating flood damage in areas with steep terrain (Nofal and van de Lindt, 2022). However, we use only flood depth as the intensity measure in this study, since it is widely used for flood loss estimation (e.g., Federal Emergency Management Agency (FEMA), 2022; Nofal and van de Lindt, 2020) and flood velocities are more difficult to record than flood depths, requiring hydraulic simulations (e.g., Kreibich et al., 2009).***


*As mentioned in section "2.4 Modelling flood vulnerability", the JRC vulnerability functions represent the best choice for this study since no specific functions have been developed for Kathmandu Valley due to a lack of available data. Note that Gentile et al. (2022) also employed the JRC vulnerability functions to assess flood risk in Kathmandu Valley. 60% is the maximum loss that occurs to brick and concrete buildings since it is assumed that the flood cannot significantly impact the structural components of the building, which represent a substantial proportion of the construction costs. This assumption is in line with that of other authors (e.g., CAPRA, 2012; FEMA, 2022) and complies with the assumptions used to develop the JRC vulnerability functions (e.g., Huizinga et al., 2017).*

*To clarify these points, we have added some lines (in **bold**) (L263-L267) to the second paragraph of section "2.4.Modeling flood vulnerability", as follows:*

*"(…) Firstly, we set maximum damage to be 100% for A and W, and 60% for all brick (BSM/BSC, RM) and concrete (i.e., RC-CCP, RC-WDS) typologies, following JRC recommendations. **The 60% maximum damage threshold used for some typologies reflects the assumption that a flood cannot damage major water-resistant structural components, which represent a substantial portion of building construction costs (Huizinga et al., 2017). This assumption is in line with other studies such as the Central American Probabilistic Risk Assessment (CAPRA) initiative (2012), which assigns 60% maximum flood losses to masonry and concrete buildings, and FEMA (2022), which indicates that major structural components are expected to withstand flood events.** (…)"*


*Thank you for this comment. To address it, we have added a description of the procedure used to classify populations as low, middle, or high-income. Briefly, we classify populations at the municipality level based on three census variables (i.e., access to mobile/telephone services, mass media communication, and means of transportation) that are treated as proxies for economic wealth.*

*In addition, as you pointed out, people from different municipalities can live in buildings of the same typologies; however, the proportions of each building typology vary per municipality. For instance, as we state in the Conclusions (L542-L544), the prevalence of unreinforced masonry buildings, which are highly vulnerable to earthquakes but not to floods (except in the case of adobe houses), is larger in low-income municipalities. Moreover, as we have now emphasized in section "2.2 Hazard modeling" (L152-L154), pluvial flooding primarily contributes to the aggregated flood hazard of the region, and our income classification results in some high-income populations living in urbanized areas more prone to pluvial*

*flooding than those that house low- and middle-income populations. A discussion of the exact reasons for the geographical and socioeconomic trends observed is not within the scope of this study.*

*This description (provided **in bold** below) has been added to the fourth paragraph of section "2.4 Modelling flood vulnerability" (L209-L308):*

*"(…) Furthermore, we use the procedure detailed in Mesta et al. (2022b) to classify populations per municipality as low, middle, or high income for facilitating socioeconomic disaggregation of financial losses. **The classification is based on three variables (i.e., access to mobile/telephone services, mass media communication, and means of transportation) recorded in the 2011 Census, which are treated as proxies for economic wealth. The Census data are aggregated at the municipality level; therefore, any variability in the population's income level within each municipality is not (and cannot be) assessed. The classification is quantile, such that the three income categories contain an equal number of municipalities. We assume that the population's income level did not vary between 2011 and 2021 and would remain unchanged in 2031, given the lack of available data to make confident projections. This assumption is at least partially supported by previous work from Cutter and Finch (2008), who suggested that the social vulnerability of a community, which is influenced by its underlying socioeconomic and demographic characteristics (e.g., income level, gender, age), is not expected to vary significantly over timeframes similar to those considered in this study."***


*Thank you for pointing this out. Indeed, in section "2.3 Modeling present and future exposure", we define seven acronyms that refer to different building typologies used in the analysis. However, as these acronyms were proposed by other authors in previous risk studies for Kathmandu Valley, we have decided not to modify them. In addition, we have removed the acronyms "GoN" (we directly use "Government of Nepal" in L77, L172) and "VDC" (which is replaced by "municipality") from the main text, but have kept the acronym "DRR" (that stands for disaster risk reduction) as this is a term widely used in risk analysis.*

*Based on your suggestions, we have added a full explanation of the building typology acronyms in Table 1 (in the table footer) and Figure 3 (please see our next response), to enhance readability. We have also added Table S1 to the Supplementary Material, containing a list of all acronyms used in the study.*

**Table 1.** Summary of considered exposure scenarios for Kathmandu Valley

| Exposure scenario | Scenario A | Scenario B | Scenario C | Scenario D |
|---|---|---|---|---|
| Year | 2021 | 2031 | 2031 | 2031 |
| Population | 3,151,741 | 3,792,232 | 3,792,232 | 3,792,232 |
| Number of buildings | 789,898 | 943,606 | 943,606 | 943,606 |
| Aggregated replacement value (€) | 17,141,921,300 | 20,309,544,200 | 20,380,261,200* 20,409,943,100** | 26,442,366,000* 26,470,663,000** |
| Building typologies featured | A, W, BSM/BSC, RC-CCP/WDS | A, W, BSM/BSC, RC-CCP/WDS | A, W, BSM/BSC, RC-CCP/WDS | RC-WDS, RM |
| DRR actions | - | - | Elevating buildings, flood-hazard-informed land-use planning | Elevating buildings, flood-hazard-informed land-use planning, structural retrofitting, building code enforcement |

A: adobe; W: wood-frame; BSM: brick stone masonry with mud mortar; BSC: brick stone masonry with cement mortar; RM: reinforced masonry; RC-CCP: current-construction-practice reinforced concrete; RC.WDS: well-designed reinforced concrete.

Note: Scenarios C and D have two different aggregated replacement values, as the cost of elevating buildings in the 100-year (*) or the 1000-year floodplains (**) differ.

**Table S1.** List of acronyms used in this work

| Acronym | Description |
|---|---|
| DRR | disaster risk reduction |
| A | adobe |
| BSM | brick/stone masonry with mud mortar |
| BSC | brick/stone masonry with cement mortar |
| W | wood-frame |
| RC-CCP | current-construction-practice reinforced concrete |
| RC-WDS | well-designed reinforced concrete |
| RM | reinforced masonry |

9. **Comment 3: Abbreviations, acronyms and numbers. Please also update the legend in Fig. 3. It takes the reader a lot of effort and search in the text to find out what the different colors actually mean.**

*We have updated Figure 5, by adding a full explanation of the building typology acronyms below the main plot.*

[Figure]

A: adobe; W: wood-frame; BSM: brick stone masonry with mud mortar; BSC: brick stone masonry with cement mortar; RM: reinforced masonry; RC-CCP: current-construction-practice reinforced concrete; RC-WDS: well-designed reinforced concrete.

**Figure 5.** Flood vulnerability functions for the different considered building typologies and their associated range of heights

10. **Comment 3: Abbreviations, acronyms and numbers. Please consider to plot the numbers in table 2. Maybe you can try simple bar or pie charts. Please also explain why scenario C and D are together in this case.**

*Thank you for your suggestions, which we have incorporated. We have added Figure 6 (bar chart) to plot the expected number of buildings in the floodplain for different flood depths (i.e., the absolute numbers previously reported in Table 2). As suggested by another reviewer, we have kept the information about the expected proportions of buildings within various depth ranges of the floodplains in Table 2.*

*Scenarios A, C, and D yield identical numbers in Figure 6 since the adoption hazard-informed land-use planning limits the expected number of buildings within the floodplain in 2031 (Scenarios C, D) to 2021 levels (Scenario A). In addition, as Scenarios C and D are associated with the same total building stock, both scenarios are grouped in Table 2 (i.e., they also have identical results in terms of the proportion of flood-exposed buildings).*

*We have modified some lines (L320-L323)  (provided **in bold** below) in the first paragraph of section "3.1 Distribution of buildings in the floodplain" to better clarify this:*

*"**Scenarios A, C, and D yield identical results in Figure 6, since the flood-hazard-informed land-use planning imposed as part of Scenarios C and D means that the expected number of buildings within the floodplain in 2031 (Scenarios C, D) remains limited to 2021 levels (Scenario A).** This measure decreases the proportion of flood-exposed buildings **in both Scenarios C and D (grouped in Table 2)** by 2.2%."*

[Figure]

**Figure 6.** Exposure to flooding: the expected number of buildings within a given range of flood depth per flooding occurrence and exposure scenario

**Table 2.** Exposure to flooding: proportions of the total building stock within a given range of flood depth per flooding occurrence and exposure scenario.

| Flooding occurrence | Scenario | 0.2-0.5 m | 0.5-1.0 m | 1.0-2.0 m | 2.0-3.0 m | 3.0-5.0 m | >5-0 m |
|---|---|---|---|---|---|---|---|
| 100-year | A | 3.8% | 3.5% | 3.8% | 1.7% | 0.9% | 0.1% |
| | B | 3.8% | 3.4% | 3.7% | 1.7% | 1.0% | 0.2% |
| | C, D | 3.1% | 2.9% | 3.2% | 1.5% | 0.8% | 0.1% |
| 1000-year | A | 3.3% | 2.9% | 3.7% | 2.1% | 2.2% | 0.8% |
| | B | 4.3% | 3.5% | 4.5% | 2.4% | 2.9% | 1.6% |
| | C, D | 2.7% | 2.4% | 3.1% | 1.7% | 1.8% | 0.7% |

11. **Comment 3: Abbreviations, acronyms and numbers. Also revise the paragraph between line 389-397. In general, you use a lot of numbers in the text. Please do not just list all the numbers, but select the numbers that you include into the text. These numbers need to underline your key findings and arguments. All the rest can go into the tables and figures, I think**

*We have made some modifications to improve the paragraph in question. In particular, we have added the numbers about the changes in mean loss ratios for Scenario B, we have reduced the text that reports the changes in mean loss ratios for Scenario C (we group municipalities in three categories instead of four), and we have removed the numbers about the changes in mean loss ratios for Scenario D. as these numbers are quite similar to those from Scenario C.*

*Also, as you suggest in another comment (please, see our response #23), we have added new maps (Figures 12 and 13) to plot the differences in loss ratios between scenarios; these maps are now mentioned in this paragraph (L452-L465):*

*"Figure 12 illustrates the absolute changes to the municipality-level mean loss ratios for the 100-year flooding occurrence, considering Scenario A as a baseline.* ***In Scenario B, the mean loss ratios show small absolute variations (between -1.0% and +1.2%) compared to Scenario A*** *since future urbanization continues occurring in both flooded and non-flooded areas. The relative effects of the building elevation strategy and the flood-hazard-informed land-use planning proposed in Scenario C are noticeable:* ***absolute reductions in mean loss ratio for Scenario C relative to Scenario A range between 2.0-2.9% in five municipalities, between 1.0-2.0% in 18 municipalities, and are less than 1.0% in the remaining 81 municipalities. The benefits of implementing additional multi-hazard DRR measures in Scenario D are almost equivalent to those in Scenario C because the seismic upgrading of the building stock does not contribute much to reducing flood risk.*** *Figure 13*

*presents the absolute changes to the municipality-level mean loss ratios for the 1000-year flooding occurrence, considering Scenario A as a baseline.* **In Scenario B, the mean loss ratios exhibit some absolute variations (between -1.0% and +2.3%) relative to Scenario A, which are larger than in the 100-year flood case; in other words, the consequences of not controlling future urbanization in flood-prone areas can increase with the severity of the considered flooding occurrence.** *The effects of the flood-specific DRR measures implemented in Scenario C are as follows:* **absolute reductions in mean loss ratio for Scenario C relative to Scenario A range between 2.0-5.6% in 9 municipalities, between 1.0-2.0% in 32 municipalities, and are less than 1.0% in the remaining 63 municipalities. The benefits of the combined DRR measures in Scenario D are comparable to those in Scenario C.**"

**Specific comments**

12. **Abstract: Make the abstract more clear. Focus on the key results, numbers and message. For me it was very difficult to get the key message of your work first time reading the abstract. Only after reading the entire article, I also understood the abstract. Clearly state the different exposure inventories and mitigation strategies you investigate. For me, your key message in very simple words is: 'Measures can reduce the risk/damage a lot. That is why we need to do it.' And you give the numbers for it. I like your first sentence in the conclusion Line 408-411. This sentence is clear to me and maybe you can use it also for the abstract.**

*Thank you for this comment. We have addressed it by including the first sentence of the Conclusions in the Abstract. We also make it clear that Scenario A corresponds to the current urban system in Kathmandu Valley, while Scenarios B, C, and D correspond to the near-future development trajectories that the valley could experience. Moreover, please note that the risk-mitigation strategies we investigate and the key results of each scenario were already clearly stated in the abstract (L15-L27); therefore, we have not made additional changes.*

*The updated Abstract (with changes marked in* **bold***) reads as follows:*

*"Flood risk is expected to increase in many regions worldwide due to rapid urbanization and climate change if adequate risk-mitigation (or climate-change-adaptation) measures are not implemented. However, the exact benefits of these measures remain unknown or inadequately quantified for potential future events in some* **flood-prone** *areas such as Kathmandu Valley, Nepal, which this paper addresses.* **This study examines the present (2021) and future (2031) flood risk in Kathmandu Valley, considering two flood-occurrence cases (with 100-year and 1000-year mean return periods)** *and using four residential exposure inventories representing the current urban system* **(Scenario A)** *or near-future development trajectories* **(Scenarios B, C, D)** *that Kathmandu Valley could experience. The* **findings reveal** *substantial mean absolute financial losses (€ 473 million and € 775 million in repair/reconstruction costs) and mean loss ratios (2.8% and 4.5%) for the respective flood-occurrence cases in current times if the building stock's quality is assumed to have remained the same as in 2011 (Scenario A). Under a "no change" pathway for 2031 (Scenario B), where the vulnerability of the expanding building stock remains the same as in 2011, mean absolute financial losses for the 100-year and 1000-year mean return period flooding occurrences would respectively increase by 16% and 14% over those of Scenario A. However, a minimum (0.20 m) elevation of existing residential buildings located in the floodplains and the implementation of flood-hazard-informed land-use planning for 2031 (Scenario C) could respectively decrease the mean absolute financial losses of the flooding occurrences by 13% and 9%, and the corresponding mean loss ratios by 27% and 23%, relative to those of Scenario A. Moreover, an additional improvement of the building stock's vulnerability that accounts for the multi-hazard-prone nature of the valley (by means of structural retrofitting and building code enforcement) for 2031 (Scenario D) would further decrease the mean loss ratios (respective reductions for the 100-year and 1000-year mean return period flooding occurrences would be 28% and 24% relative to those of Scenario A). The largest mean loss ratios computed in the four scenarios are consistently associated*

*with populations of the highest incomes, largely located in the floodplains. In contrast, the most significant benefits of risk mitigation (i.e., largest reduction in mean absolute financial losses or mean loss ratios between scenarios) are experienced by populations of the lowest incomes. This paper's main findings can inform decision makers about the benefits of investing in forward-looking multi-hazard risk-mitigation efforts."*

**13. introduction: Needs to be more concise. There are a lot of general information. Please try to tailor it to the specific content of your article. There is lot about climate change, but in this study you do not assess climate change impacts. This is almost a bit misleading.**

*Thank you for this comment, which we have addressed by making the following modifications to the Introduction:*

- *In the fourth paragraph (L63-L81), we have removed four lines that provided specific details of the climate change scenarios developed by the Government of Nepal. The fourth paragraph (with changes marked **in bold**) now reads as follows:*

  *"The 2017 Terai flood and earlier major events have emphasized the significant risk that flooding continuously imposes on the Nepalese population. While flood risk is already substantial, several ongoing trends in the country could further amplify this risk in the coming years. **Firstly,** Nepal is projected to be one of the fastest urbanizing countries in the world over the 2018-2050 period (United Nations, 2019b), which could lead to significantly larger amounts of flood exposure. While urban growth is gaining pace across different regions of Nepal, Kathmandu Valley represents the "hub" of urban development in the country (Timsina et al., 2020). A previous study by the authors (Mesta et al., 2022b) revealed that urban land in Kathmandu Valley could reach 352 km2 in 2050, almost doubling its current size and covering half the total valley extent. A significant share of this new urbanization is projected to occupy the valley's most hazardous (at least in terms of flooding and liquefaction) and socially-vulnerable regions (Mesta et al., 2022b). **Secondly,** other natural hazards such as earthquakes have unveiled the poor state of Nepal's building stock and physical infrastructure, which is caused by a combination of low-quality building materials, deficient construction practices, low compliance with building codes, as well as aging, and deterioration (Bothara et al., 2018; Varum et al., 2018). Traditional materials, such as bamboo/wood, stone, and mud, are still preferred in many regions of the country (especially in rural areas) due to their availability and low cost (Bothara et al., 2018). However, buildings made of bamboo/wood or mud suffer severely from flood damage (e.g., Becker Andrea B. et al., 2011; Fatemi et al., 2020) due to low durability and high permeability. **Thirdly, climate change scenarios developed by the Government of Nepal (Ministry of Forests and Environment, 2019) reveal a rising trend in precipitation (for all seasons, except the pre-monsoon season) in the medium term (2016-2045) and long-term (2036-2065).** Therefore, it is critical to determine the potential benefits of implementing disaster risk reduction (DRR) strategies in the country (particularly Kathmandu Valley) towards preventing devastating economic losses and casualties in future major natural hazard events. "*

- *In the fifth paragraph, we have added a sentence (in **bold**) (L97) to emphasize that the impact of climate change is not within the scope of this work:*

  *"(…) The methodology employs a scenario-based flood loss estimation approach, using 100-year and 1000-year mean return period flood occurrence maps and four potential present (2021) and future (2031) exposure and vulnerability scenarios, focusing only on residential buildings. **Note that the impact of climate change is not explicitly considered within this work.** (…)"*

**14. Page 1 Line 11: 'multi-hazard-prone area': There are multiple important hazard in the area, but you do not assess multi-hazards, as far as I understood.**

*Yes, this study only focuses on assessing flood risk. We have replaced "multi-hazard-prone area" with "flood-prone area" (L11), to address this comment.*

**15. Page 1 Line 14: Be careful with the word 'predict'. Maybe better use 'Our results hint/point at/suggest…'**

*We have replaced "the results predict" with "the findings reveal" (L15).*

**16. Page 2 Line 56-57: Many readers are not familiar with these locations. Please explain where and what 'Terai regions'.**

*We have added some lines (in **bold**) (L59-L60) to the paragraph in question, to provide more information about the Terai regions:*

*"(…) **Terai is one of Nepal's three ecological belts (together with Mountain, and Hill), and covers the alluvial and fertile plains along the southern part of the country (Government of Nepal, 2017). (…)"***

*Thank you for your suggestion. We have addressed it by adding a new figure (Figure 1), which provides an overview of this study's flood risk modeling approach.*

[Figure]

**Figure 1.** Overview of the flood risk modeling approach used in this study

**22. Figure 4: Wouldn't it be better to (also) show the absolute numbers? In my opinion, showing the percentage without information of building density can be a bit missleading. Like this, it does not provide good information on the the spatial distribution of flooded buildings, I think. Can you maybe plot the buildings on the map directly?**

*Thank you for this comment. We have replaced the previous maps (Figures 4 and 5) with Figures 7 and 8. In these new maps, we plot both the expected number of buildings in the floodplain and the percentage of buildings in the floodplain.*

*Also, as suggested by another reviewer, we have rescaled the pie charts that show the distribution of the expected number of buildings within various depth ranges of the floodplains. Figures S1 and S2 (in the Supplementary Material) contain the updated pie charts.*

*We clarify here that we have developed our exposure model by downscaling municipality-level exposure data to 30m resolution urban maps of built-up areas in the valley (see Figure 3), which are too coarse to contain exact building footprints; this is a typical feature of regional risk exposure models. Thus, we cannot directly plot the buildings on the map, as requested. To clarify this in the manuscript, we have added some lines (in **bold**) (L202-L203, L206-L207) to sections "2.3.1 Scenario A (population and buildings for 2021)" and "2.3.2 Scenarios B, C, D (population and buildings for 2031)":*

*L202-L203: "(…) **We disaggregate the exposure data to match the 30 m spatial resolution of the urban map containing the 2021 built-up areas (see Figure 3)."***

*L206-L207: "(…) **We disaggregate the exposure data into the urban map containing the 2021 and 2031 built-up areas (see Figure 3)."***

[Figure]

**Figure 7.** Spatial distribution of buildings in the 100-year floodplain for a) Scenario A, b) Scenario B, and c) Scenarios C and D. The numbers plotted inside each municipality correspond to the expected number of buildings in the floodplain.

[Figure]

**Figure 8**. Spatial distribution of buildings in the 1000-year floodplain for a) Scenario A, b) Scenario B, and c) Scenarios C and D. The numbers plotted inside each municipality correspond to the expected number of buildings in the floodplain.

**23. Figure 5: I am not sure the mean loss ratio on a municipality level is a very interesting thing to plot here. As you have the exact flood maps, why not plot damage using the inundation maps. In this way, hotspots of damage are visible. Please also try to calculate difference maps, e.g. between A and C to show the benefit of certain measures.**

*Thank you for pointing this out. We have decided to keep our municipality-level loss ratio maps in the manuscript, as we believe these maps provide a broad overview of flood risk hotspots.*

*As explained in the previous comment, we clarify that we have developed our exposure model by downscaling municipality-level exposure data to 30m resolution urban maps of built-up areas in the valley (see Figure 3), which are too coarse to contain exact building footprints. Thus, we cannot directly plot building-level damage on the map, as requested.*

*Based on your comment, we have added Figures 12 and 13 that show how mean loss ratios change in Scenarios B, C, and D relative to Scenario A. These maps are used in the discussion on the benefits of risk-mitigation measures, provided in the last paragraph (L452-L465) of section "3.2 Losses":*

*"Figure 12 illustrates the absolute changes to the municipality-level mean loss ratios for the 100-year flooding occurrence, considering Scenario A as a baseline. In Scenario B, the mean loss ratios show*

*small absolute variations (between -1.0% and +1.2%) compared to Scenario A, since future urbanization continues occurring in both flooded and non-flooded areas. The relative effects of the building elevation strategy and the flood-hazard-informed land-use planning proposed in Scenario C are noticeable: absolute reduction in mean loss ratios range between 2.0-2.9% in five municipalities, between 1.0-2.0% in 18 municipalities, and is less than 1.0% in the remaining 81 municipalities, relative to Scenario A. The benefits of implementing additional multi-hazard DRR measures in Scenario D are quite similar to those in Scenario C because the seismic upgrading of the building stock does not contribute much to reducing flood risk. Figure 13 presents the absolute changes to the municipality-level mean loss ratios for the 1000-year flooding occurrence, considering Scenario A as a baseline. In Scenario B, the mean loss ratios exhibit some absolute variations (between -1.0% and +2.3%) relative to Scenario A, which are larger than in the 100-year flood case; in other words, the consequences of not controlling future urbanization in flood-prone areas are worse for a more severe flooding occurrence. The effects of the flood-specific DRR measures implemented in Scenario C are as follows: absolute reduction in mean loss ratios range between 2.0-5.6% in 9 municipalities, between 1.0-2.0% in 32 municipalities, and is less than 1.0% in the remaining 63 municipalities, relative to Scenario A. The benefits of the combined DRR measures in Scenario D are also comparable to those in Scenario C."*

[Figure]

**Figure 12.** Absolute changes to the municipality-level mean loss ratios for a) Scenario B; b) Scenario C; c) Scenario D, relative to Scenario A, for the 100-year flooding occurrence,

[Figure]

**Figure 13.** Absolute changes to the municipality-level mean loss ratios for a) Scenario B; b) Scenario C; c) Scenario D, relative to Scenario A, for the 1000-year flooding occurrence.

**24. Line 450: Data availability: This is not sufficient, I think. Are there ownership issues and you cannot provide the data sets? Is it possible to put the data into a FAIR repository? At least the core data and an example data set?**

*Thank you for this comment. Indeed, it is important to point out that the hazard and urban map datasets are available online through public repositories. We have updated the section "Data availability" by providing links to access the datasets used in this study:*

*"Data availability. The flood hazard maps are available online from the METEOR project (available at https://maps.meteor-project.org/map/flood-npl/, last accessed December 2022). The urban maps for Kathmandu Valley are available online through a public repository (available at https://doi.org/10.5281/zenodo.7406981, last accessed December 2022). Other datasets used and/or analyzed during the current study are available from the corresponding author on reasonable request."*

---

## Author Comment (AC2)

**REVIEWERS' COMMENTS TO THE AUTHOR:**

**Reviewer 2:**

**General comments**

1. The manuscript "Quantifying the potential benefits of risk-mitigation strategies on future flood losses in Kathmandu Valley, Nepal" addresses flood risk under four scenarios of urbanization and climate change (Scenarios A-D) with a focus on a multi-hazard prone area by computing the associated mean absolute financial losses and mean loss ratios. I believe the manuscript could represent a substantial contribution to the understanding of flood events and especially their consequences and therefore fits perfectly the special issue "Estimating and predicting natural hazards and vulnerabilities in the Himalayan region". Nevertheless, before the manuscript is considered for publication, the authors need to address some concerns.

*Many thanks for your overall positive assessment of our manuscript and your insightful comments, which are addressed in detail below.*

2. The introduction provides valuable information about the relevance of flood events and risk assessment in the region of study. However, I believe the authors need to provide some additional details about the selected methodology (LL 96-98)). Why has this specific methodology been chosen? I recommend the authors to justify this selection. Is it based on previous works? Please add the associated references.

*The methodology used in this study represents a conventional approach to flood risk assessment involving the modeling of hazard, exposure, and vulnerability components. Further details of each step of the methodology are provided in section "2. Material and methods" and in Figure 1.*

*We have replaced the term "integrates" with "is" in the sentence in question (L95) to avoid misinterpretation:*

*"The methodology **is** a scenario-based flood loss estimation approach, using 100-year and 1000-year mean return period flood occurrence maps and four potential present (2021) and future (2031) exposure and vulnerability scenarios, focusing only on residential buildings."*

3. I believe a figure showing the framework of the work with a step-by-step diagram will be very useful for the readers to better understand the methods implemented and highlight the scope of the work

*Thank you for this suggestion. We have incorporated it by adding Figure 1, which provides an overview of the flood risk modeling approach used in this study.*

[Figure]

**Figure 1.** Overview of the flood risk modeling approach used in this study

**4. Regarding the information provided in Table 2, I recommend the authors to include a graph with the "expected number of buildings exposed to flooding" (Y axis) for the different scenarios (represented with colors for example) and the different flood depth (X axis), instead of the overwhelming Table 2. The authors could keep the information about the percentage in Table 2.**

*Thank you for this suggestion, which we have incorporated. We have added Figure 6, which contains plots of the expected number of buildings in the floodplain. We have kept the information about the expected proportions of buildings within various depth ranges of the floodplains in Table 2.*

[Figure]

**Figure 6.** Exposure to flooding: the expected number of buildings within a given range of flood depth, per flooding occurrence and exposure scenario

**Table 2.** Exposure to flooding: proportions of the total building stock within a given range of flood depth, per flooding occurrence and exposure scenario.

| Flooding occurrence | Scenario | 0.2-0.5 m | 0.5-1.0 m | 1.0-2.0 m | 2.0-3.0 m | 3.0-5.0 m | >5-0 m |
|---|---|---|---|---|---|---|---|
| 100-year | A | 3.8% | 3.5% | 3.8% | 1.7% | 0.9% | 0.1% |
| | B | 3.8% | 3.4% | 3.7% | 1.7% | 1.0% | 0.2% |
| | C, D | 3.1% | 2.9% | 3.2% | 1.5% | 0.8% | 0.1% |
| 1000-year | A | 3.3% | 2.9% | 3.7% | 2.1% | 2.2% | 0.8% |
| | B | 4.3% | 3.5% | 4.5% | 2.4% | 2.9% | 1.6% |
| | C, D | 2.7% | 2.4% | 3.1% | 1.7% | 1.8% | 0.7% |

**5. It is unfortunate that in Figure 4 and Figure 5 the regions with 1% or 10% buildings in the floodplain are represented by a very small pie chart. This makes very difficult the interpretation. Is there any way that the authors could rescale these charts?**

*Thanks for this comment. We have rescaled the pie charts to a uniform size to improve their interpretation. However, we have moved these maps (Figures S1 and S2) to the Supplementary Material, since the accompanying text in the last paragraphs of section "3.1 Distribution of building in the floodplain" (L340-L355) is more focused on the percentage of buildings in the floodplain than the prevalence of different flood depths at municipality level.*

*In addition, as suggested by another reviewer, we have replaced Figures 4 and 5 with Figures 7 and 8, which contain plots of the expected number of buildings in the floodplain and the percentage of buildings in the floodplain.*

[Figure]

**Figure S1.** Expected proportions of buildings within various depth ranges of the 100-year floodplain for a) Scenarios A, C, D, and b) Scenario B.

Scenarios A, C, and D yield identical results, since the flood-hazard-informed land-use planning imposed as part of Scenarios C and D means that the expected number of buildings within the floodplain in 2031 (Scenarios C, D) remains limited to 2021 levels (Scenario A).

[Figure]

**Figure S2.** Expected proportions of buildings within various depth ranges of the 1000-year floodplain for a) Scenarios A, C, D, and b) Scenario B.

Scenarios A, C, and D yield identical results, since the flood-hazard-informed land-use planning imposed as part of Scenarios C and D means that the expected number of buildings within the floodplain in 2031 (Scenarios C, D) remains limited to 2021 levels (Scenario A).

[Figure]

**Figure 7.** Spatial distribution of buildings in the 100-year floodplain for a) Scenario A, b) Scenario B, and c) Scenarios C and D. The numbers plotted inside each municipality correspond to the expected number of buildings in the floodplain.

[Figure]

**Figure 8**. Spatial distribution of buildings in the 1000-year floodplain for a) Scenario A, b) Scenario B, and c) Scenarios C and D. The numbers plotted inside each municipality correspond to the expected number of buildings in the floodplain.

**6. Table 3 shows relevant information. However, the authors have a very detailed amount of data that could be used to have a more complete table. Could the authors include the absolute values for the different districts? And income levels per district for example?**

*Thank you for pointing this out. We attempted to incorporate the absolute results disaggregated by districts and income into Table 3, as suggested. The results of this attempt are provided in the following table, which we believe contains too much information for one table. Most importantly, the purpose of Table 3 is to provide a general overview of the changes in flood risk for the entire Kathmandu Valley, which is in line with how we discuss the results in the main text (L406-L442). Therefore, we have decided to keep Table 3 in its current format.*

*However, we have added new Tables S2 and S3 to the Supplementary Material, where we provide the changes in mean absolute financial losses and mean loss ratios disaggregated by district and income level, respectively.*

**Table 3.** Mean loss metrics for Scenario A, and absolute changes to these metrics in Scenario B, C, and D. The mean absolute financial losses are disaggregated by district and income level. **(this table is not included in the manuscript)**

| Flooding occurrence | Metric | District/ income level | Scenario A | Scenario B | Scenario C | Scenario D |
|---|---|---|---|---|---|---|
| 100-year | Mean absolute financial losses (€) | **Total** | **472,932,965** | **+ 74,654,044** | **-63,283,799** | **+ 52,691,045** |
| | | Bhaktapur | 62,253,439 | +8,617,871 | -6,306,202 | +9,605,264 |
| | | Kathmandu | 345,346,326 | +57,348,232 | -51,220,080 | +32,270,581 |
| | | Lalitpur | 65,333,200 | +8,687,941 | -5,757,517 | +10,815,200 |
| | | Low income | 38,592,710 | +18,097,384 | -6,774,800 | +2,259,014 |
| | | Middle income | 103,189,196 | +23,146,134 | -11,400,874 | +14,191,417 |
| | | High income | 331,151,059 | +33,410,526 | -45,108,125 | +36,240,614 |
| | Mean loss ratio | **Total** | **2.8%** | **-0.06%** | **-0.75%** | **-0.77%** |
| 1000-year | Mean absolute financial losses (€) | **Total** | **774,793,163** | **+ 107,901,808** | **-66,393,767** | **+ 130,500,162** |
| | | Bhaktapur | 89,654,403 | +12,377,766 | -5,258,360 | +18,393,467 |
| | | Kathmandu | 576,162,142 | +83,341,425 | -56,444,738 | +88,865,127 |
| | | Lalitpur | 108,976,618 | +12,182,617 | -4,690,669 | +23,241,568 |
| | | Low income | 60,564,250 | +25,800,358 | -7,288,996 | +7,639,070 |
| | | Middle income | 162,179,628 | +35,446,740 | -11,381,018 | +29,947,118 |
| | | High income | 552,049,285 | +46,654,710 | -47,723,753 | +92,913,974 |
| | Mean loss ratio | **Total** | **4.5%** | **-0.17%** | **-1.0%** | **-1.1%** |

**Table S2.** Mean loss metrics for Scenario A, and absolute changes to these metrics in Scenarios B, C, and D, disaggregated by district

| Flooding occurrence | Metric | District | Scenario A | Scenario B | Scenario C | Scenario D |
|---|---|---|---|---|---|---|
| 100-year | Mean absolute financial losses (€) | Bhaktapur | 62,253,439 | +8,617,871 | -6,306,202 | +9,605,264 |
| | | Kathmandu | 345,346,326 | +57,348,232 | -51,220,080 | +32,270,581 |
| | | Lalitpur | 65,333,200 | +8,687,941 | -5,757,517 | +10,815,200 |
| | Mean loss ratio | Bhaktapur | 3.5% | -0.09% | -0.81% | -0.84% |
| | | Kathmandu | 2.7% | -0.07% | -0.80% | -0.82% |
| | | Lalitpur | 2.4% | -0.02% | -0.49% | -0.52% |
| 1000-year | Mean absolute financial losses (€) | Bhaktapur | 89,654,403 | +12,377,766 | -5,258,360 | +18,393,467 |
| | | Kathmandu | 576,162,142 | +83,341,425 | -56,444,738 | +88,865,127 |
| | | Lalitpur | 108,976,618 | +12,182,617 | -4,690,669 | +23,241,568 |
| | Mean loss ratio | Bhaktapur | 5.0% | -0.13% | -1.0% | -1.0% |
| | | Kathmandu | 4.6% | -0.20% | -1.1% | -1.2% |
| | | Lalitpur | 4.0% | -0.11% | -0.66% | -0.74% |

**Table S3.** Mean loss metrics for Scenario A, and absolute changes to these metrics in Scenarios B, C, and D, disaggregated by income level

| Flooding occurrence | Metric | Income level | Scenario A | Scenario B | Scenario C | Scenario D |
|---|---|---|---|---|---|---|
| 100-year | Mean absolute financial losses (€) | Low | 38,592,710 | +18,097,384 | -6,774,800 | +2,259,014 |
| | | Middle | 103,189,196 | +23,146,134 | -11,400,874 | +14,191,417 |
| | | High | 331,151,059 | +33,410,526 | -45,108,125 | +36,240,614 |
| | Mean loss ratio | Low | 2.1% | +0.10% | -0.88% | -0.90% |
| | | Middle | 2.4% | -0.08% | -0.71% | -0.73% |
| | | High | 3.0% | -0.04% | -0.69% | -0.72% |
| 1000-year | Mean absolute financial losses (€) | Low | 60,564,250 | +25,800,358 | -7,288,996 | +7,639,070 |
| | | Middle | 162,179,628 | +35,446,740 | -11,381,018 | +29,947,118 |
| | | High | 552,049,285 | +46,654,710 | -47,723,753 | +92,913,974 |
| | Mean loss ratio | Low | 3.3% | +0.06% | -1.3% | -1.3% |
| | | Middle | 3.7% | -0.14% | -1.0% | -1.0% |
| | | High | 5.0% | -0.15% | -0.94% | -1.0% |

7. **Section 3 is in my opinion much more oriented to describing the results rather than a discussion of the obtained results. I believe the authors could benefit from adding a specific discussion section to go one step further and try to find the reasoning behind the obtained results. Furthermore, the authors could answer critical questions such as: what is the impact or correlation of the flood depth on the losses? Are there any thresholds that could be established based on the present results in terms of flood depth leading to specific losses? Why are high income levels suffering the highest losses? How are the flooding risk areas differing from other hazards such as earthquakes? Are there any solutions that would be beneficial to prevent simultaneously both hazards? Additionally, from figures 7 and 8 the authors could integrate an interesting discussion about measures planning, prioritizing high risk areas and highlight the benefits of taking action.**

*Some of these questions were already addressed in various sections of the manuscript. For instance, in both the Results and Conclusions, we discuss the relationship between loss and income (L375-L381; L537-L539), the benefits of taking action (L420-L430), and the varying effects that risk-mitigation measures have for flood and earthquake hazards (L541-L550). Solutions that simultaneously prevent losses from floods and earthquakes are those that we propose in Scenario D (L218-L222). Moreover, it is the vulnerability functions themselves that correlate flood depth and loss, and provide information on threshold depths that lead to certain losses. We attempted to make broad municipality-level associations between mean water depth and mean loss ratios (or absolute losses); however, as the proportion of flooded buildings and building typologies vary per municipality, we could not find any interesting general trend.*

*L375-381: "From Figure 9 (panel b), we identify some variability in the mean loss ratios by income level. All scenarios produce the highest mean loss ratios for the high-income population, which reflects their disproportionate share of buildings in inundated areas (there are also minor differences in the prevalence of building typologies between income groups, but the three income groups are dominated by brick and concrete typologies). For instance, in Scenario A, the proportion of buildings in the floodplain is 15% in high-income municipalities, 11% in middle-income municipalities, and 12% in low-income municipalities. Proportions of buildings that experience flood depths below and above 2.0 m, respectively, are 81%-19% for high-income municipalities, 75%-25% for middle-income municipalities, and 84%-16% for low-income municipalities."*

*L537-L539: "Second, this study reveals that high-income populations are exposed to the highest mean loss ratios across both flooding-occurrence cases due to having the largest proportions of buildings in the floodplain."*

*L420-L430: "In contrast, the relative decrease in mean absolute financial losses for Scenario C shows that, despite a growing population, elevating existing buildings and implementing flood-hazard-informed land-use planning could significantly reduce flood losses in the future. However, it should be noted that risk-mitigation actions implemented in Scenario C would still leave the building stock highly vulnerable to earthquakes and thus do not completely address multi-hazard risk in the valley, which is left to Scenario D. The relative increase in mean absolute financial losses in Scenario D is associated with the larger replacement value of its building stock (due to the structural retrofitting and building code enforcement measures implemented), highlighting a tension between short-term (pre-hazard occurrence) costs and long-term benefits (i.e., after the occurrence of hazard events) associated with holistic DRR measures. In summary, Scenario D demonstrates that, despite a growing population, adequate DRR measures that aim to improve the building stock's quality (for better sustaining both flood and earthquake damage) as well as incentivize urbanization away from flood-sensitive areas can limit (but not reduce) flood losses in the future."*

*L541-550: "This discrepancy illustrates that risk-mitigation measures can have varying effects for different hazards; therefore, DRR plans should be appropriately tailored for a specific region or sub-region and account for multiple hazards. Kathmandu Valley's building stock is highly vulnerable to earthquakes due to the prevalence of URM buildings (particularly in low-income municipalities), such as adobe and brick/stone masonry. However, this feature of the building stock does not make it particularly susceptible to flood damage (except in the case of adobe houses, which are made of mud), which is why a multi-hazard approach to DRR that also considers earthquake vulnerability strengthening measures has little effect on the mean loss ratios (and even results in increased mean absolute financial losses) in this study. Instead, the flood risk is mainly controlled by the extent to which populations are located in the floodplain. Considering that hazard intensities vary spatially and that flooding and earthquake-induced ground shaking can affect different proportions of buildings in a given municipality, combinations of individual DRR measures should be investigated to find the optimal DRR solution for a given municipality."*

*L218-L222: "Scenario D incorporates DRR measures that account for the multi-hazard-prone nature of Kathmandu Valley and can reduce both flood and seismic risk effectively. This means that it still includes the Scenario D structural retrofitting policies and building code enforcement seismic risk-mitigation interventions introduced in Mesta et al. (2022a) (i.e., A, BSM, BSC building typologies are replaced by RM; the RC-CCP typology is converted to RC-WDS), in addition to the flood-related DRR measures proposed in Scenario C."*

**8. I strongly recommend adding in the discussion a section about the limitations of the study. Along the manuscript many limitations and simplifications have been mentioned (e.g.: maps resolution, neglection of urbanization effects on flood hazards, basement consideration, random association of number of stories, component-level vulnerability information not available), please discuss the implications of all these aspects and the associated uncertainties for the findings of this study. What are the most impactful simplifications? The authors suggest addressing specific limitations in the future in the conclusions, but these statements need a previous proper discussion about the impact of these limitations on the accuracy and uncertainty of the results of the present work.**

*We have addressed this comment by adding a new section "Discussion", where we comment on the main limitations of this study. This new section (L483-L527) reads as follows:*

*"**4 Discussion***

[revised manuscript text omitted]


*Thank you for pointing this out. We place the sentence* "However, the primary purpose of this study is to test different exposure/vulnerability scenarios using a common flood hazard input that is open and easily accessible; developing bespoke fine-resolution flood hazard models for the study area is not within the scope of this work." *at the end of the paragraph because the two previous sentences describe two different limitations of our hazard model. Therefore, we have kept the sentence in question in its current position (L136-L139).*

*Note that the effects of urbanization on flood hazard are described in the Introduction (L44-L48). We have made modifications (marked in* **bold**) *to the sentence in question (L134-L136) to also clarify it there:*

*L134-L136:* "In addition, urbanization effects on flood hazard **(i.e., the replacement of natural ground with impermeable surfaces, changes to drainage or irrigation systems, and deforestation can increase runoff during precipitation events)** are not explicitly accounted for by the Fathom-Global model and are therefore neglected in our analyses."

*L44-L48:* "Specifically, rapid urbanization – which is expected to mainly feature across cities in Asia and Africa over the next few decades (United Nations, 2019a) – could increase flood exposure and vulnerability (e.g., Hemmati et al., 2020) and intensify flood hazard (by increasing runoff during precipitation events, due to the replacement of natural ground with impermeable surfaces, changes to drainage or irrigation systems, and deforestation, for instance), if not correctly managed."

**11. L130: Please add a reference for the sentence in brackets.**

*We have added references to the sentence in question (L142-L143):*

"Decision makers frequently use this type of map (e.g., to identify flood risk zones in the United States) (Ludy and Kondolf, 2012; Federal Emergency Management Agency (FEMA), 2010)."

*For the figures in question, it is important to note that the captions already provide a short contextual description at the start that refers to both subplots. For instance, in the caption "Figure 4. Fluvial-pluvial flood maps for a) 100-year mean return period; and b) 1000-year mean return period flooding occurrences", the description "Fluvial-pluvial flood maps for" applies to both subplots a) and b). Thus, we do not believe that additional descriptions are necessary.*

---

## Author Response (AR2)

**Reviewer 1:**

1. I congratulate the authors for the improvement of the quality of the manuscript "Quantifying the potential benefits of risk-mitigation strategies on future flood losses in Kathmandu Valley, Nepal". The authors addressed most of the comments and this is reflected in the clarity and coherence of the manuscript. The introduction is much clearer and highlights the research gaps the study is covering. The new figures and sections, such as the study area and the discussion section provide a more appropriate structure. Nevertheless, I have some final remarks that could be considered as moderate or minor comments.

*Many thanks for your overall positive assessment of our manuscript and your insightful additional comments, which are addressed in detail below.*

2. Comment 1: I believe the added figures (and the new updated versions of previous figures) help the reader to better understand the context and the key messages of the study. However, according to the new figures I would suggest to slightly restructure the sections as follows:

   2. Study area (instead of 2.1). In this section Figure 2 and Figure 3 would become Fig. 1 and Fig. 2. I recommend the authors to include some additional information about the basins' properties, climate and flood typology or references where this information is available.

   3. Materials and methods (instead of 2). I believe the figure (currently Figure 1) showing the overview of the flood risk modeling approach used in this study is really helpful. In my opinion, it is more appropriate for the manuscript readability if it is followed directly by the subsections that constitute the boxes/buckets of the scheme (i.e.: Hazard modeling, Modeling present and future exposure, Modeling flood vulnerability). Please update LL 113-114, section numbers, and Figure numbers if the manuscript is restructured as suggested.

*Thank you for your comment, which we have addressed by making the following modifications to the manuscript:*

- *We have restructured the sections as per your suggestions (i.e., Section 2 is now "Study area", Section 3 is now "Materials and methods", etc.) and updated the numbering of the sections and figures accordingly.*

- *We have added more information about the Bagmati river basin (marked in **bold**) to the first paragraph of section "2. Study area" (L101-106), as follows:*

*"This study focuses on Kathmandu Valley, Nepal, which is surrounded by the Himalayan mountains and lies within the Bagmati river basin. **The Bagmati river is 170 km in length, originates north of Kathmandu Valley at an altitude of 2690 m, and flows south through Nepal to reach the Ganges in India. Climatically, the Bagmati river basin can be divided into three regions: subtropical climate (elevations lower than 1000 m), warm temperate climate (elevations between 1000 m and 2000 m), and cool temperate climate (elevations higher than 2000 m; Dhital et al., 2013). The annual average and monsoon average rainfall***

*of its catchment area are 1800 mm and 1500 mm respectively, and the mean temperature varies between 10°C and 30°C (Dhital et al., 2013)."*

*References:*
1. *Dhital, Y. P., Tang, Q., and Shi, J.: Hydroclimatological changes in the Bagmati River Basin, Nepal, J. Geogr. Sci., 23, 612–626, 2013.*

- *We have updated Figure 3 by renaming the three main steps included in the scheme to match the name of the sections used in the main text. Each step name now also specifies the corresponding section number.*

[Figure]

**Figure 3.** Overview of the flood risk modeling approach used in this study

3. **Comment 2: The results section provides a comprehensive view of the affected buildings located in the floodplain for the four scenarios, and the associated losses per district and income level. Although the authors explain the reason why scenario D is associated to higher mean absolute financial losses, I am wondering in the authors could provide any comparative between scenarios C and D regarding earthquake impacts. This could be based on literature and past studies. I believe it is relevant to highlight the benefits of scenario D to give some context for the readers.**

*Thank you for your comment. To address it, we have added a comparison of Scenarios C and D in terms of potential earthquake impacts for Kathmandu Valley (Mesta et al., 2022a), which*

*highlights the benefits of the multi-hazard DRR plan proposed in Scenario D. This additional discussion (in **bold**) (L422-427) is within section "4.2 Losses", and reads as follows:*

*"(…) However, it should be noted that risk-mitigation actions implemented in Scenario C would still leave the building stock highly vulnerable to earthquakes, and thus do not completely address multi-hazard risk in the valley, which is left to Scenario D. **Note that a previous study by the authors (Mesta et al., 2022a) revealed that not implementing seismic risk-mitigation actions for the valley (i.e., analogous to Scenario C in this study) could increase mean absolute financial seismic losses in the future (2031) by more than € 1.7 billion (+20%) relative to equivalent current levels. In contrast, improving the seismic strength of buildings (i.e., similar to Scenario D in this study), could reduce mean absolute financial seismic losses in the future by more than € 1.1 billion (-14%) relative to equivalent current levels.** The relative increase in mean absolute financial losses in Scenario D is associated with the larger replacement value of its building stock (due to the structural retrofitting and building code enforcement measures implemented). This highlights a tension between short-term up-front costs (incurred before the occurrence of hazard events) and long-term benefits (after the occurrence of hazard events) associated with holistic DRR measures. In summary, Scenario D demonstrates that, despite a growing population, adequate DRR measures that aim to improve the building stock's quality (for better sustaining both flood and earthquake damage) as well as incentivize urbanization away from flood-sensitive areas can limit (but not reduce) mean absolute financial flood losses in the future."*

*References:*
1. *Mesta, C., Cremen, G., and Galasso, C.: Quantifying the Potential Benefits of Risk-Mitigation Strategies on Present and Future Seismic Losses in Kathmandu Valley, Nepal, Earthq. Spectra, 2022a.*

4. **Comment 3: The discussion section gives a better idea of the interpretation of the results obtained in the present study. Additionally, listing and commenting the limitations of the study is key to link the outputs to strategies for decision-makers and to highlight the scope of the work and suggest the direction of future studies. However, I missed a paragraph focusing on the obtained results and how these results could help decision-makers. For example, could we extract guidelines for prioritizing action in regions under higher risk? Why are some regions showing much higher positive impacts of DDR measures? Do these regions share any common characteristics that could lead to a step-wise action plan? Or is the present analysis meant to display which regions deserve a more local assessment with a proper analysis of costs? This information would support the statements included in the conclusions (e.g.: LL595-597, LL606-609).**

*This is a great comment. We have incorporated your suggestions within an additional final paragraph of the Discussion section, which reads as follows:*

*L537-L540: "The results obtained in this study provide valuable information for decision makers about drivers of exacerbated future flood risk and can help to support appropriate policy making. The proposed framework could also inform high-level guidelines for identifying potential flood risk hotspots that deserve a more detailed local DRR assessment (e.g., including higher-resolution data/models, a proper analysis of costs, a tailored analysis of DRR measures, etc.)."*

*Also, note that a detailed discussion of the results is already provided in the Conclusions section.*

**5. Minor comments: L319 Please delete "at least", since it is implied when using "partially".**

*We have removed "at least" from the sentence.*

**6. L530 Delete "which is now discussed".**

*We have removed "which is now discussed" from the sentence.*

**7. L569 Instead of "the exact values of absolute losses are not of particular interest or relevance", I would recommend to say "the uncertainty associated to the absolute losses is not within the scope of this study".**

*We have replaced "the exact values of absolute losses are not of particular interest or relevance" with "uncertainty associated with the absolute losses is not within the scope of this study".*

**Reviewer 2:**

1. **General comment. Thank you for revising the manuscript. Text, figures as well as discussion improved substantially, in my opinion. In the following, a small list of minor comments:**

*Many thanks for your overall positive assessment of our manuscript and your insightful additional comments, which are addressed in detail below.*

2. **I find the abstract a bit difficult to read. There are a lot of numbers in the abstract. Maybe it is possible to summarize the findings more beyond the numbers and present the results in a more general and overarching way? Some key numbers certainly are good, but with all the scenarios it is a bit too detailed, I think.**

*Thank you for your comment. To improve the readability of the abstract, we have shortened the sentences showing the results for Scenarios B, C, and D by removing the references to each flood occurrence (e.g., we have replaced "(…) mean absolute financial losses for the 100-year and 1000-year mean return period flooding occurrences would respectively increase by 16% and 14% over those of Scenario A" with "(…) mean absolute financial losses would increase by 14%-16% over those of Scenario A.").*

*We have kept the key results of each scenario in the abstract. Note that we provide the results in terms of mean absolute financial losses and mean loss ratios for Scenario A, but we only provide the relative variations for Scenarios B, C, and D.*

*The updated Abstract (with changes marked in **bold**) reads as follows:*

*"**Abstract.** Flood risk is expected to increase in many regions worldwide due to rapid urbanization and climate change if adequate risk-mitigation (or climate-change-adaptation) measures are not implemented. However, the exact benefits of these measures remain unknown or inadequately quantified for potential future events in some flood-prone areas such as Kathmandu Valley, Nepal, which this paper addresses. This study examines the present (2021) and future (2031) flood risk in Kathmandu Valley, considering two flood-occurrence cases (with 100-year and 1000-year mean return periods) and using four residential exposure inventories representing the current urban system (Scenario A) or near-future development trajectories (Scenarios B, C, D) that Kathmandu Valley could experience. The findings reveal substantial mean absolute financial losses (€ 473 million and € 775 million in repair/reconstruction costs) and mean loss ratios (2.8% and 4.5%) for the respective flood-occurrence cases in current times if the building stock's quality is assumed to have remained the same as in 2011 (Scenario A). Under a "no change" pathway for 2031 (Scenario B), where the vulnerability of the expanding building stock remains the same as in 2011**, mean absolute financial losses would increase by 14%-16% over those of Scenario A**. However, a minimum (0.20 m) elevation of existing residential buildings located in the floodplains and the implementation of flood-hazard-informed land-use planning for 2031 (Scenario C) could **decrease the mean absolute financial losses by 9%-13% and the mean loss ratios by 23%-27%**, relative to those of Scenario A. Moreover, an additional improvement of the building stock's vulnerability that accounts for the multi-hazard-prone nature of the valley (by means of structural retrofitting and building code enforcement) for 2031 (Scenario D) would further **decrease the mean loss ratios by 24%-28% relative to those of Scenario A**. The*

*largest mean loss ratios computed in the four scenarios are consistently associated with populations of the highest incomes, which are largely located in the floodplains. In contrast, the most significant benefits of risk mitigation (i.e., largest reduction in mean absolute financial losses or mean loss ratios between scenarios) are experienced by populations of the lowest incomes. This paper's main findings can inform decision makers about the benefits of investing in forward-looking multi-hazard risk-mitigation efforts."*

**3. Line 105: It reads a bit as if you use four potential present and four potential future scenarios. So in total eight. Maybe rephrase here to avoid any misunderstandings.**

*Thank you for your comment. We have rephrased the sentence in question (with changes marked in **bold**) to avoid any misunderstanding:*

*L94-L95: "The methodology is a scenario-based flood loss estimation approach, using 100-year and 1000-year mean return period flood occurrence maps and **four exposure and vulnerability scenarios representing the current (2021) and potential near-future (2031) development trajectories for the valley**, focusing only on residential buildings."*

**4. The river network in Fig. 2 looks a bit weird with the cutoff stream segments. Maybe you can try to find another data source or delineate the river network yourself?**

*Thank you for your comment. Data on the river network comes from OpenStreetMap (OSM), which is the best available source of information for the river network in Kathmandu Valley. OSM information for small streams is incomplete, which explains why some stream segments appear to be cut off in Figure 2. We have added a note in the figure caption to clarify this.*

[Figure]

**Figure 1.** Physical map of Kathmandu Valley. The river network is taken directly from OpenStreetMap (OSM); small streams appear cut off where OSM data are incomplete. Inset map data: © Google Earth.

**5. Line 136: Add version number of the Fathon-Global model.**

*Thank you for your comment. We have specified the version number (i.e., 2.0) in the sentence.*

**6. Line 154: What do you mean by 'current flood' hazard'? Isn't also the 1000-year flood current flood hazard? Please avoid to phrase as if the 100-year flood is the present and the 1000-year flood the future flood.**

*We have removed the phrase* "and is intended to represent current flood hazard" *from the sentence, to avoid any misunderstanding.*

**7. Fig. 12 and 13 a) Those maps are a bit confusing, I think. You show that the mean loss ratio goes down despite increases amount of flooded and damaged building. It goes down because it is the mean ration and there are a lot of new buildings outside the flooded areas, right? So even if there are more damaged buildings the mean loss ration goes down. In scenario B more buildings get flooded and damaged, so it would be better to show this in a map, I think. Maybe you can try to compare the difference of the number of buildings affected or the total damage?**

*Thank you for your comment. Indeed, mean loss ratios in some municipalities decrease from Scenario B to Scenario A where future urbanization outside the floodplain is larger than that in it.*

*To clarify this point, we have added some lines (marked in **bold**) to the eighth paragraph of section "4.2 Losses", as follows:*

*L461-L463:* *"In Scenario B, the mean loss ratios show small absolute variations (between -1.0% and +1.2%) compared to Scenario A, since future urbanization continues to occur in both flooded and non-flooded areas.* ***Some municipalities experience a decrease in mean loss ratio (see Figure 7), where future urbanization outside the floodplain is larger than that within it.****"*

*Moreover, we greatly appreciate your suggestion to include a new map in the manuscript showing differences in the number of buildings affected by floods. However, we have decided not to incorporate it because Figures 7 and 8 (section 4.1 "Distribution of buildings in the floodplain") already provide the number of buildings in the floodplain for each scenario; differences between the scenarios can be derived from these maps, so mapping them explicitly would unnecessarily increase the manuscript's length.*